# Hydrogen-deuterium exchange mass spectrometry captures distinct dynamics upon substrate and inhibitor binding to a transporter

Ruyu Jia [1,4], Chloe Martens [1,3,4 ✉], Mrinal Shekhar [2], Shashank Pant [2], Grant A. Pellowe [1], Andy M. Lau [1], Heather E. Findlay[1], Nicola J. Harris [1], Emad Tajkhorshid [2], Paula J. Booth[1 ✉] & Argyris Politis [1 ✉]

Proton-coupled transporters use transmembrane proton gradients to power active transport of nutrients inside the cell. High-resolution structures often fail to capture the coupling between proton and ligand binding, and conformational changes associated with transport. We combine HDX-MS with mutagenesis and MD simulations to dissect the molecular mechanism of the prototypical transporter XylE. We show that protonation of a conserved aspartate triggers conformational transition from outward-facing to inward-facing state. This transition only occurs in the presence of substrate xylose, while the inhibitor glucose locks the transporter in the outward-facing state. MD simulations corroborate the experiments by showing that only the combination of protonation and xylose binding, and not glucose, sets up the transporter for conformational switch. Overall, we demonstrate the unique ability of HDX-MS to distinguish between the conformational dynamics of inhibitor and substrate binding, and show that a specific allosteric coupling between substrate binding and protonation is a key step to initiate transport.

[1] Department of Chemistry, King's College London, 7 Trinity Street, London SE1 1DB, UK. [2] Center for Biophysics and Quantitative Biology, Department of Biochemistry, NIH Center for Macromolecular Modeling and Bioinformatics, Beckman Institute for Advanced Science and Technology, University of Illinois at Urbana-Champaign, Urbana, IL, USA. [3] Present address: Center for Structural Biology and Bioinformatics, Universite Libre de Bruxelles, Brussels, Belgium. [4] These authors contributed equally: Ruyu Jia, Chloe Martens. ✉email: chloe.martens@ulb.be; paula.booth@kcl.ac.uk; argyris.politis@kcl.ac.uk

Structural biology of membrane proteins has evolved at an increasing pace over the past few years[1]. The more high-resolution structural information becomes available, the clearer it appears that complementary dynamic information is required to understand the mechanism of a protein of interest[2]. Energy coupling in secondary transporters is a good example of the type of information that static structures cannot directly provide about molecular mechanisms[3]. Although it is clear that these transporters alternate between different conformations ranging from open to the cytoplasm (inward facing, IF) to open to the extracellular medium (outward facing, OF), the molecular chain of events leading to these transitions are difficult to capture[4]. Specifically, the identification of the allosteric networks linking ion and substrate binding, and the ensuing protein conformational changes, are hard to deduce from structural snapshots[5]. Thus, linking structure to mechanism at a molecular level requires characterizing the conformational dynamics of membrane proteins[6].

Among the techniques available to study conformational changes, hydrogen-deuterium exchange mass spectrometry (HDX-MS) is a newcomer for the study of membrane proteins[7]. This technique reports on the exchange of amide hydrogens on the protein backbone in the presence of deuterated solvent at a peptide level of resolution[8]. The main advantage over more established methods such as Förster resonance energy transfer (FRET) and Double Electron Electron Resonance (DEER) is that it does not require covalent labelling of the protein of interest, thus bypassing a lot of the molecular biology work and controls[9]. The method also requires lower amount of sample compared to other biophysical methods (such as nuclear agnetic resonance or X-ray crystallography)[10] and tolerates sample heterogeneity and complexity[11,12]. Hydrogen/deuterium (H/D) exchange, however, does not strictly report on distance changes involved in conformational transitions. Rather, it reports on the stability of H-bond of the amide backbone, which is mainly conditioned by two parameters: local structural dynamics and solvent accessibility[13,14]. We have shown previously that for a series of transporters, the changes in solvent accessibility can be correlated with conformational changes in most cases[15]. This is particularly helpful to understand the molecular mechanism of transporters as they switch between OF and IF conformations[16]. The conformational effect of ligand binding, mutation of conserved residues, or both, can be tested in a systematic way by comparing the H/D exchange pattern in different conditions, in so-called differential (Δ) HDX-MS experiments. Assuming that no major changes in the backbone stability of transporters occur when introducing either the ligand or a mutation, ΔHDX-MS offers a quick and easy readout of the conformational transition between different states.

The symporter XylE of the ubiquitous Major Facilitator Superfamily (MFS) family is a bacterial homologue of human glucose transporters GLUTs 1–4[17], with a sequence similarity of ~50%. The *xyle* gene was first isolated in 1987[18]. The expressed protein was shown to use the proton-motive force to catalyse xylose translocation across the membrane of *Escherichia coli*[19]. The majority of bacterial sugar transporters relies on ion gradients to energize transport[20]. Mammalian GLUTs transporters in contrast are facilitators. This difference in sugar transport energetics between human and bacteria appears to arise from the scarcer availability of sugar for bacteria, compared to humans whose sugar levels in the blood are in the mM range[21]. Despite this difference, several residues and structural motifs are strictly conserved from XylE to GLUTs 1–4, critical either for substrate recognition or to enable structural rearrangements[19]. The crystal structure of XylE has been solved in multiple conformations: inward-open, inward-occluded and outward-occluded with substrate xylose and inhibitor glucose bound[17,19,22]. The structures of the xylose-bound and glucose-bound protein are virtually identical, with only minor differences in the interaction network at the binding site[17]. This observation raises questions on how the transporter discriminates between substrate and inhibitor and how the potential differences are translated into conformational changes. Despite advances, the coupling between xylose and proton binding and conformational changes are not understood. Two transmembrane acidic residues located away from the binding pocket are likely candidates for the protonation step: D27 on helix 1 and E206 on helix 6[23–25]. Biochemical assays have identified D27 as a critical component for active transport, with mutations at this site abolishing function[17,22,23]. Neighbouring residue E206 has been suggested to play a role in modulating the p$K_a$ of D27, to regulate its ability to bind and release proton[22,24]. Binding assays carried out on wild-type (WT)[17,24] and D27N mutant[23,24] show that they both bind xylose with a similar affinity ($K_d$ 0.3 mM).

In a previous study, we carried out an extensive characterization of the conformational dynamics of XylE by HDX-MS, to establish the mechanistic role of a conserved network of charged residues located on the intracellular side[11]. For benchmarking purposes, we locked the transporter in an OF conformation by replacing a conserved glycine necessary for the structural transition by a bulky tryptophan. This work provided a set of ΔHDX maps associated with transitions toward either the IF or OF states and allowed us to identify regions of the protein that can be used as conformational reporters. Peptic peptides from these regions are used as a fingerprint to guide interpretation of the ΔHDX experiments performed in the present study. Representative peptides from this benchmark experiment are provided in Supplementary Fig. 1. Here we performed HDX-MS measurements of the proton-coupled symporter XylE in the presence of its substrate xylose, inhibitor glucose and mutations at candidate protonation sites D27 and E206. The systematic HDX analysis coupled to molecular dynamics (MD) simulations identifies differences in structural dynamics and allosteric events between xylose and glucose binding, providing a rationale for inhibitor vs. substrate distinction.

## Results

To dissect the role of proton and substrate binding, all the possible combinations between WT and mutants mimicking protonation—D27N, E206Q and E206Q&D27N—in the apo- and substrate-bound states were tested (Fig. 1a, b). At least three biological triplicates were used for each ΔHDX-MS experiment comparing two different protein states, as recommended for this type of experiments[26]. Heat maps of Relative Fractional Uptake per amino acid and Woods plots showing peptides with significant ΔHDX are available as Supplementary Figs. S2 and S3. As the peptides generated by enzymatic digestion can be different between biological replicates, we used Deuteros[27,28] to identify peptides showing a significant difference (confidence interval of 99%) in deuterium uptake for each individual ΔHDX-MS experiment and carried out an extra step of curating the data to represent only the peptides that are present in two or more of all replicates. Peptides containing the mutation(s) were excluded from analysis. It is noted that sequence coverage of >90% was obtained in most cases (Supplementary Fig. S3), allowing us to monitor the dynamics of nearly the entire protein.

**Protonation of D27 controls the conformational transition.** We first set out to understand the effect of protonation on the dynamics of XylE in the absence of substrate or inhibitor. To this end, we carried out ΔHDX-MS experiments comparing the WT

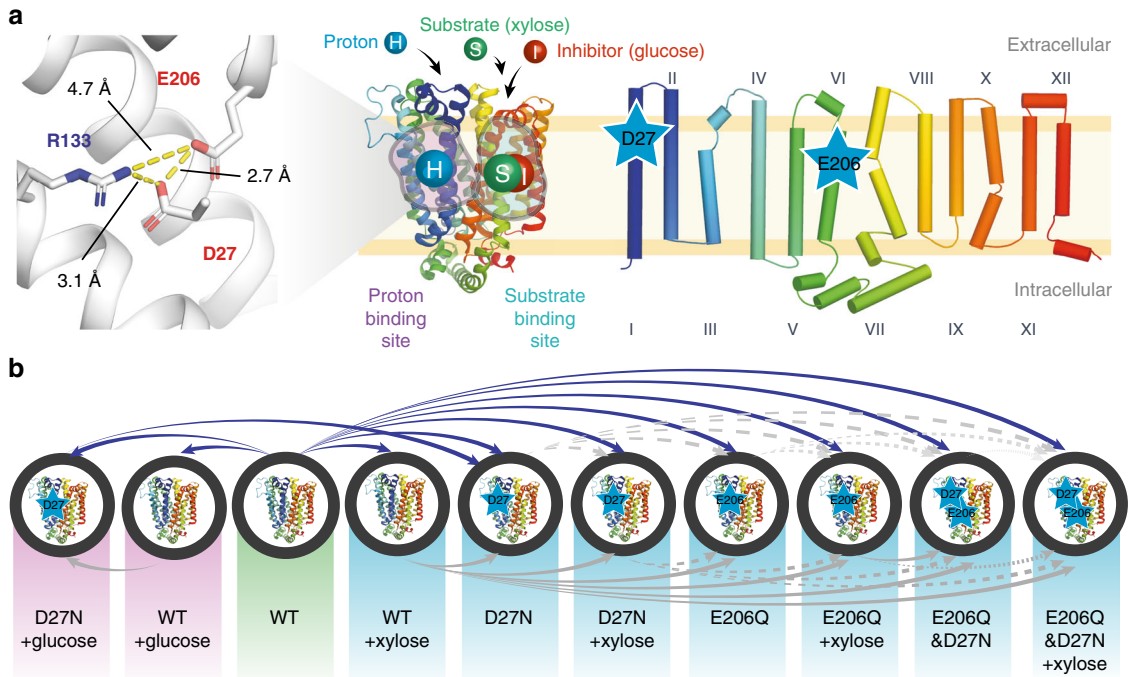

**Fig. 1 Structure of XylE and proposed experiment design. a** Topological and 3D structure of XylE (PDB: 4GBY)[17]. Three charged residues of interest in the proton binding site (D27, R133 and E206) are shown with their intra-residue distances. **b** Twenty-eight combinations of eight different protein states of XylE WT and three mutants (D27N, E206Q and E206Q&D27N) in the presence or absence of a substrate (xylose) and four combinations of XylE WT and mutant D27N in the presence or absence of an inhibitor (glucose) were studied in this work. All structural representations were generated using PyMol. Mutated residues are indicated by a star. Tables reporting experimental details for each ΔHDX-MS experiment are available as Supplementary Data File 1.

protein with the mutants. We observe that protonation mimics D27N and E206Q cause an overall decrease in deuterium uptake on both the extracellular and intracellular sides compared to the WT protein (Fig. 2a). No significant exchange is observed in the transmembrane regions, which are mostly solvent inaccessible. Interestingly, the double-mutant E206Q&D27N shows a decrease of deuterium uptake on the extracellular side coupled to an increase on the intracellular side—this corresponds to a ΔHDX pattern typical for the transition of transporter toward an IF state (Fig. 2b)[11]. To understand the sequence of events enabling protein transition to the IF state, we carried out ΔHDX-MS experiments comparing the single to double mutants. By comparing the double-mutant E206Q&D27N to single-mutant E206Q, we found ΔHDX pattern typical of a transition toward an IF state (Fig. 2c). By contrast the ΔHDX of E206Q&D27N vs. D27N only showed minor differences in deuterium uptake (Fig. 2d). Taken together, these results suggest that D27 protonation is the main driver of the conformational transition to IF state, as long as a proton is already present on E206. To confirm that the ΔHDX observed in our experiments was the result of conformational changes and not changes in global stability caused by the mutations, we performed thermal unfolding experiments, monitored by circular dichroism (CD) measurements under temperature gradient[29,30]. A decrease in global stability often stems from an increase in local unfolding events, which in turn affects H-bond stability, thereby leading to an increase in H/D unrelated to an OF/IF conformational transition[31]. No significant change in stability was observed between the WT and the mutants below 50 °C (Supplementary Fig. S4), which comforted us that the changes observed with HDX were mainly caused by conformational changes.

**Substrate or inhibitor binding favours the OF state**. We then investigated the role of the substrate xylose and inhibitor glucose on the conformational equilibrium of XylE. The protein and

mutants were first incubated with 750 μM of xylose and the effect was followed by HDX-MS. The comparison between the protein in the presence and absence of xylose consistently shows that the presence of the substrate leads to a ΔHDX pattern typical of a transition toward an OF conformation, an increase in deuterium uptake on the extracellular side coupled to a decrease in deuterium uptake on the intracellular side (Fig. 3a and Supplementary Fig. S5). We performed similar experiments with the inhibitor glucose (750 μM). We observed that glucose also stabilizes the OF conformation, regardless of the presence of mutations (Fig. 3e and Supplementary Fig. S5). We thus observe a systematic shift of the conformational equilibrium toward the OF state in the presence of either xylose or glucose. This transition is observed for all XylE variants, suggesting that substrate binding favours the OF conformation regardless of the prior protonation state of D27 or E206 and the apo conformational ensemble of the transporter. These results are in line with the observed OF states of the ligand-bound structures captured by X-ray crystallography[17]. However, this raises the question about how the transition of the loaded transporter toward the IF conformation occurs.

**Allosteric coupling between D27 protonation and substrate binding**. Next, we went on to characterize how the combined effect of substrate binding and protonation mimics have an impact on the conformational dynamics, to emulate a fully loaded transporter. We carried out ΔHDX-MS experiments of the mutant proteins vs. the WT, in the presence of xylose. Strikingly, we observed that D27N vs. WT in the presence of xylose (Fig. 4a) presented a different ΔHDX pattern compared to the apo experiment (Fig. 2a). The mutation leads to an increase in deuterium uptake on both sides of the protein, a pattern different from all the other ΔHDX patterns observed so far. This increased uptake on both sides of the transporter suggests that there is a decrease in H-bond stability on the entire protein, suggesting

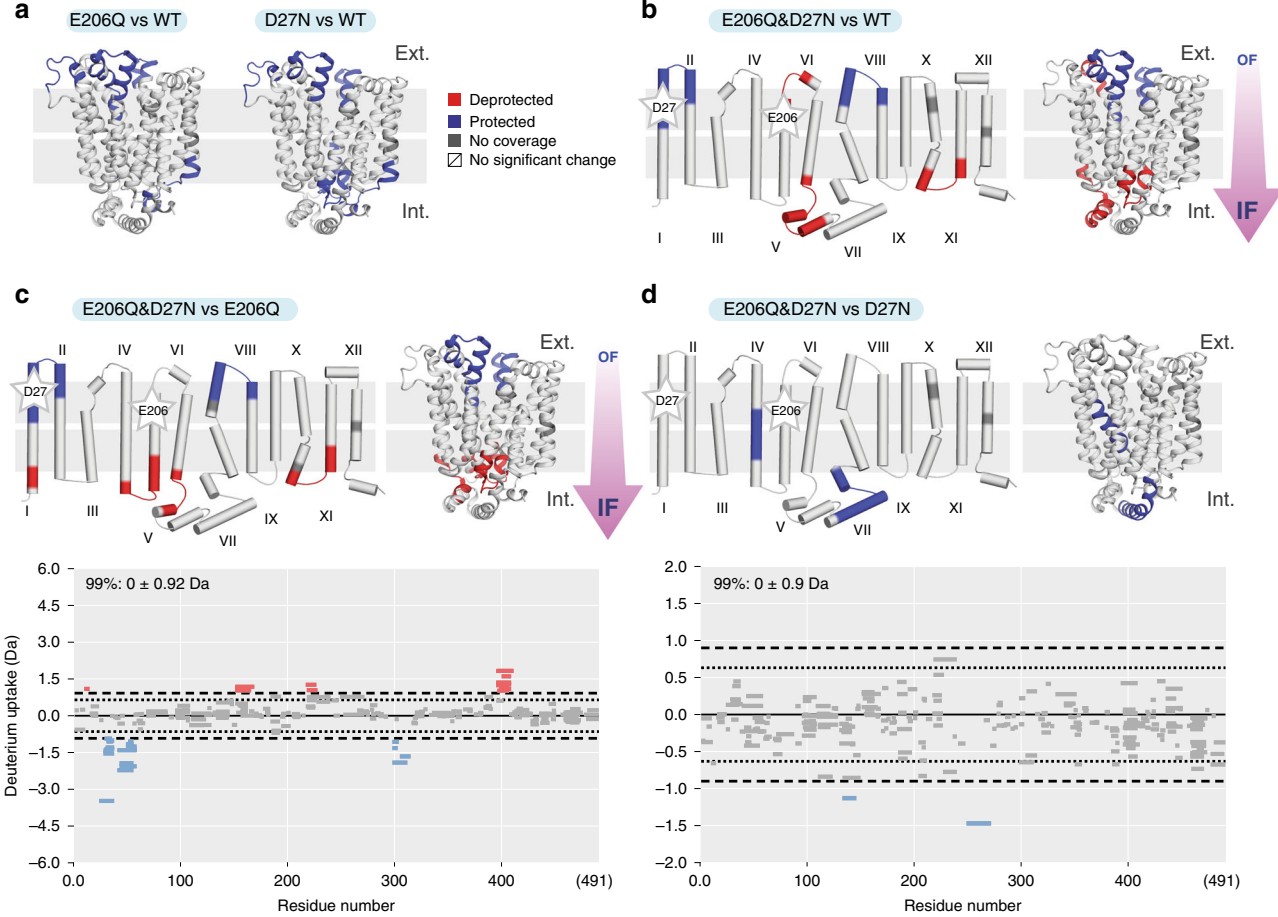

**Fig. 2 Conformational change of XylE through mutation (protonation mimic). a** Differential uptake pattern (ΔHDX) map comparing XylE E206Q or D27N mutants to the WT. **b** ΔHDX map between XylE E206Q&D27N and WT state. **c** ΔHDX map and woods plot between XylE E206Q&D27N and XylE E206Q. **d** ΔHDX map and Woods plot between XylE E206Q&D27N and XylE D27N. Each bar on Woods plots represents a single peptide with peptide length indicated by the bar length. Figures are projected onto topological and 3D protein structure (PDB: 4GBY) using PyMol. Blue and red regions indicate a negative (protected) and a positive (deprotected) deuterium uptake difference, respectively. Mutated residues are indicated by a star.

that the combined presence of the mutation and the substrate leads to increased conformational heterogeneity. We hypothesize that D27N + xylose compared to D27N alone favours transition-competent conformations, where transition refers to the conformational change that allows the transporter to move between the OF and IF states. A similar ΔHDX pattern was observed when comparing D27N minus E206Q, the double-mutant E206Q&D27N minus E206Q, but not E206Q minus WT, suggesting that the coupling between substrate binding and protonation is specific to D27 (Supplementary Fig. S6). We then performed the same experiment comparing D27N with the WT in the presence of the inhibitor glucose. To our surprise, this time we observed a pattern consistent with an OF conformation, suggesting that glucose binding tips the conformational equilibrium even more toward the OF state (Fig. 4b). This comparison between xylose and glucose indicates that only a bona fide substrate can lead to the conformational transition. Overall, our results suggest that D27 protonation is the trigger for conformational cycling of the protein, while protonation of E206 has little effect. The shift toward a "transition-competent conformational ensemble" demonstrates that a specific allosteric coupling exists between the mutation/protonation of D27 and binding of the substrate xylose.

**MD simulations suggest protonation leads to substrate destabilization.** To understand the allosteric interplay between D27

protonation state and xylose binding, we ran all-atoms MD simulations on the ligand-bound and apo structures. We calculated the intrinsic $pK_a$ values of the residues D27 and E206 in the crystal structures using PROPKA[32]. The $pK_a$ of D27 ranges from 4.35 to 3.64, and that of E206 from 11 to 12.13 depending on the conformation in which the protein was crystallized (Supplementary Fig. S7). The intrinsic $pK_a$ values of these residues suggest that in the conformations captured in the crystal structures, XylE is protonated at E206 and deprotonated at D27.

We performed MD simulations of XylE embedded in a 1-palmitoyl-2-oleoyl phosphatidylethanolamine (POPE) lipid bilayer with the residue D27, either unprotonated or protonated and E206 always protonated, using either the xylose-bound or glucose-bound structure. We clearly observe that in the case of unprotonated D27, xylose remains stably bound, essentially retaining the crystal structure pose (Fig. 5a, i-ii). In contrast, xylose adopts multiple rotameric states in D27-protonated state (Fig. 5b, i-ii), suggesting that xylose-binding stability is conditional on the absence of a proton on D27. Furthermore, instability of xylose is facilitated by the increased solvation of the substrate-binding site (Fig. 5b, iii). In contrast, the glucose-bound simulation with D27 protonated retains the crystallographic pose with essentially a similar pattern of substrate stability and solvation as the xylose-bound D27-unprotonated state (Fig. 5c).

By looking closer at the effect of solvation of the subtract pathway and binding site, we observed that TM1 (bearing residue

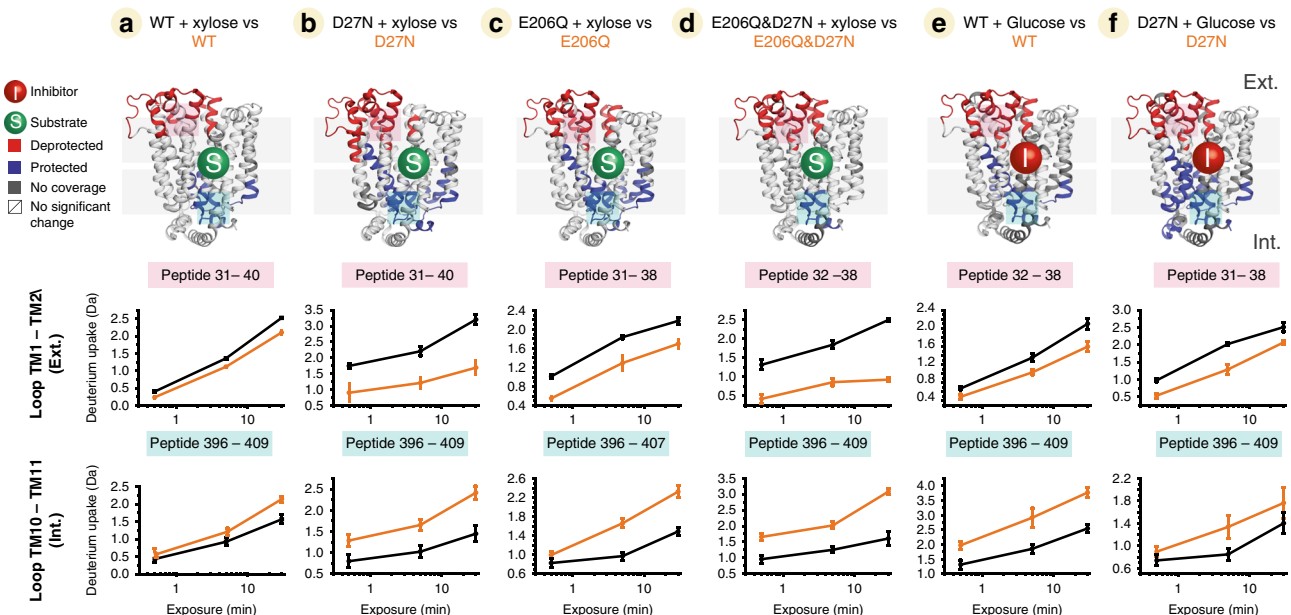

**Fig. 3 Conformational change of XylE through substrate/inhibitor binding. a** Differential deuterium uptake pattern and peptide deuterium uptake plots (peptide on TM1-TM2 loop and on TM10-TM11 loop) between **a** WT with substrate (xylose)-bound state and apo state, and **b** between D27N with substrate (xylose)-bound state and D27N state. **c** E206Q with substrate (xylose)-bound state and E206Q state, **d** E206Q&D27N with substrate (xylose)-bound state and E206Q&D27N state and **e** WT with substrate (xylose)-bound state and apo state. **f** D27N with inhibitor (glucose)-bound state and D27N state. Figures are plotted onto 3D protein structure (PDB: 4GBY). Blue and red regions indicate negative (protected) or positive (deprotected) deuterium uptake differences between the aforementioned states, respectively. SDs for each time point are plotted as error bars (n = 3).

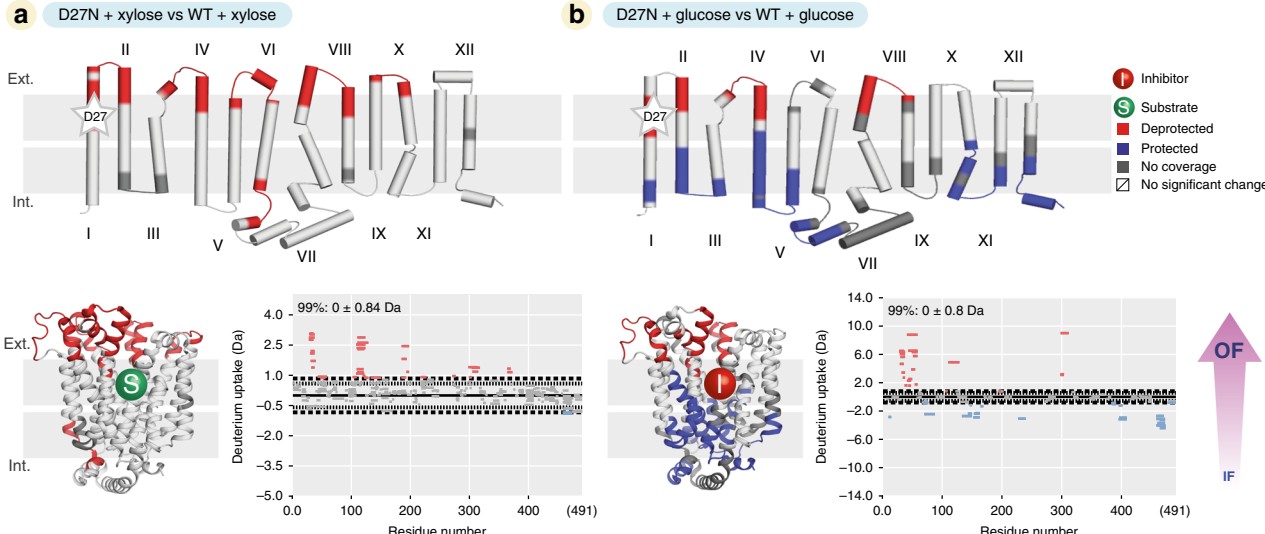

**Fig. 4 Conformational transition of XylE through combined effect of substrate binding and mutation. a** ΔHDX map and Woods plot between XylE D27N with XylE WT in the presence of substrate (xylose). **b** ΔHDX map and Woods plot between XylE D27N and XylE WT in the presence of inhibitor (glucose).

D27) undergoes a decrease in helical content that is markedly lower in the presence of xylose and protonated D27, compared to either the xylose-bound/D27-unprotonated or glucose-bound/D27-protonated cases (Fig. 6a). This decrease in helicity is correlated with the increased solvation: the water-mediated bonds between TM1 and specific and ordered water molecules disappear as more water molecules come in. Consequently, residue T28 (next to D27) reorients its methyl group toward the binding site, resulting in an overall decrease of TM1 helicity that propagates and amplifies along the extracellular side of the helix (Supplementary Fig. S8). Furthermore, the importance of TM1 flexibility in regulating the conformational transition is corroborated by the

observation of a similar loss in TM1 helicity (residues I31-G33) for XylE in IF-occluded (PDB: 4JA3) and IF-open (PDB: 4JA4) states. To identify the molecular mechanisms leading to such differences between glucose and xylose binding, we carried out a detailed analysis of residue rearrangements happening at the sugar-binding site. The dihedral angles of the residues involved in substrate binding were calculated (Supplementary Fig. S9). Only minor differences between xylose- and glucose-bound states were observed with the notable exception of residues N294 and Q168. Both residues are pointing away from xylose (Fig. 6b) but towards glucose (Fig. 6c). In contrast, these residues point away from the ligand-binding site in XylE in IF conformation.

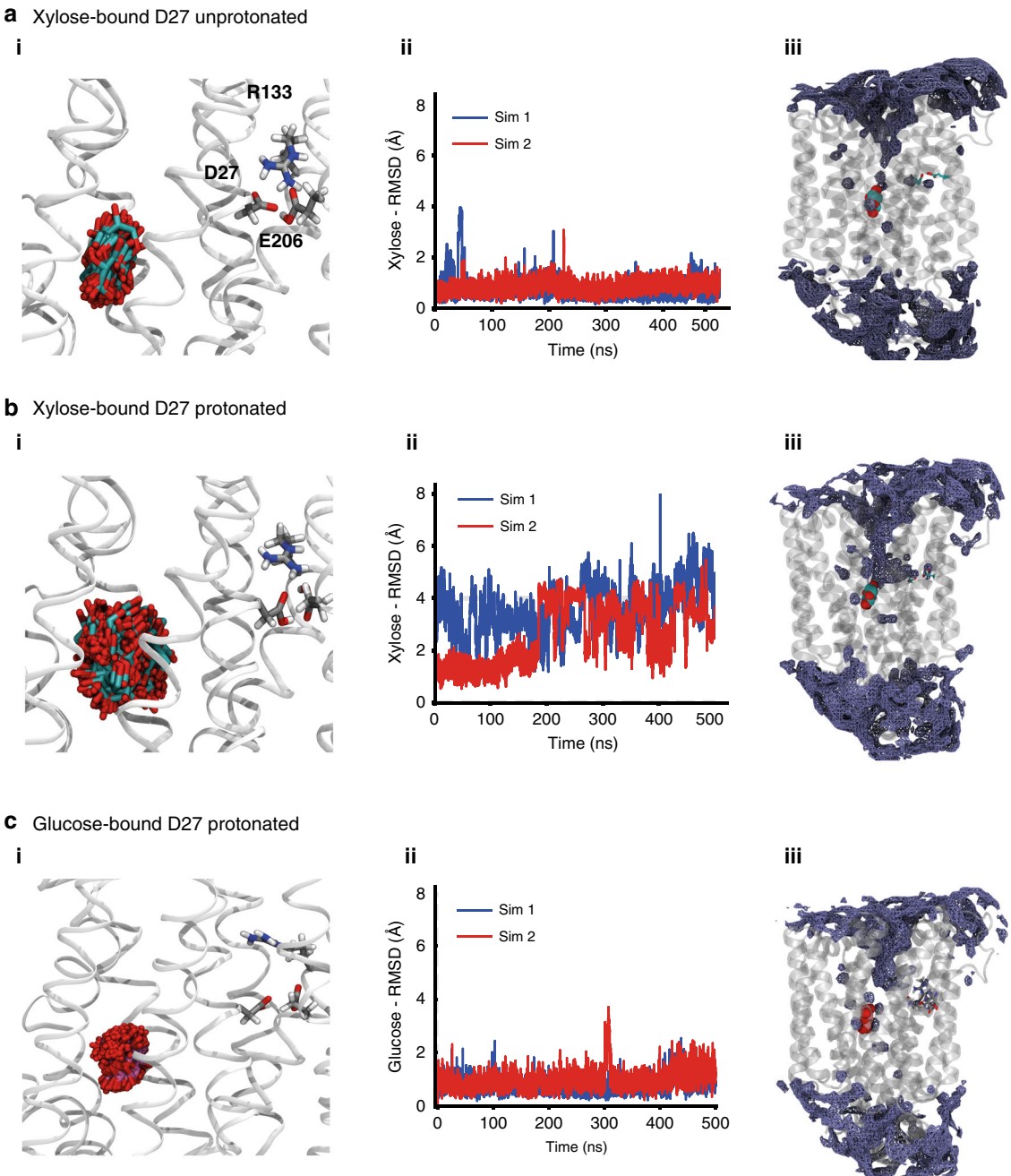

**Fig. 5 Molecular dynamics simulations reveal differences in ligand stability and water accessibility. a** (i, ii) MD simulations of xylose (shown in cyan and red sticks)-bound XylE (initiated from the crystal structure pose, PDB: 4GBY) in the D27-unprotonated state highlights that the bound substrate remains stable through the two independent 500 ns-long MD simulations. **a** (iii) Water density depicted as average occupancy of water molecules during 500 ns of MD simulation show that the water accessibility to the binding site is restricted. Ligand in the crystal structure (PDB: 4GBY) conformation is depicted for reference. **b** (i, ii) MD simulations (initiated from the crystal structure pose, PDB: 4GBY) of xylose bound to protonated D27 shows an increase in substrate flexibility and in **b** (**iii**) water accessibility to the binding site, depicted as average occupancy over 500 ns MD simulation. Ligand in crystal structure (PDB: 4GBY) pose depicted reference. **c** (i, ii) Bound glucose molecule (shown in pink and red sticks) in D27-protonated XylE retains the crystallographic pose with a similar pattern of substrate stability in MD simulations initiated from the crystal structure (PDB: 4GBZ). **c** (iii) Solvation as the xylose-bound unprotonated state. The stability was characterized by monitoring the RMSD of the xylose or glucose with respect to the crystal structure pose.

The MD predictions corroborate the HDX-MS results at several levels. First, the instability of xylose binding and the increase of water molecules along the substrate pathway observed upon D27 protonation matches with the global increase in H/D exchange observed. Second, the high calculated $pK_a$ value of E206 suggests that this residue is protonated most of the time during HDX-MS experiments carried out at pH 7.0. This explains why E206Q mutation leads to minor or no changes in ΔHDX-MS experiments carried out in the presence of a substrate ((Fig. 4a and Supplementary Fig. S6). Third, the simulations confirm that the coupling between D27 and substrate-binding site strictly depends on xylose binding, whereas glucose binding does not lead to increased solvation and loss of secondary structure of helix 1. The combination of MD predictions and HDX-MS results

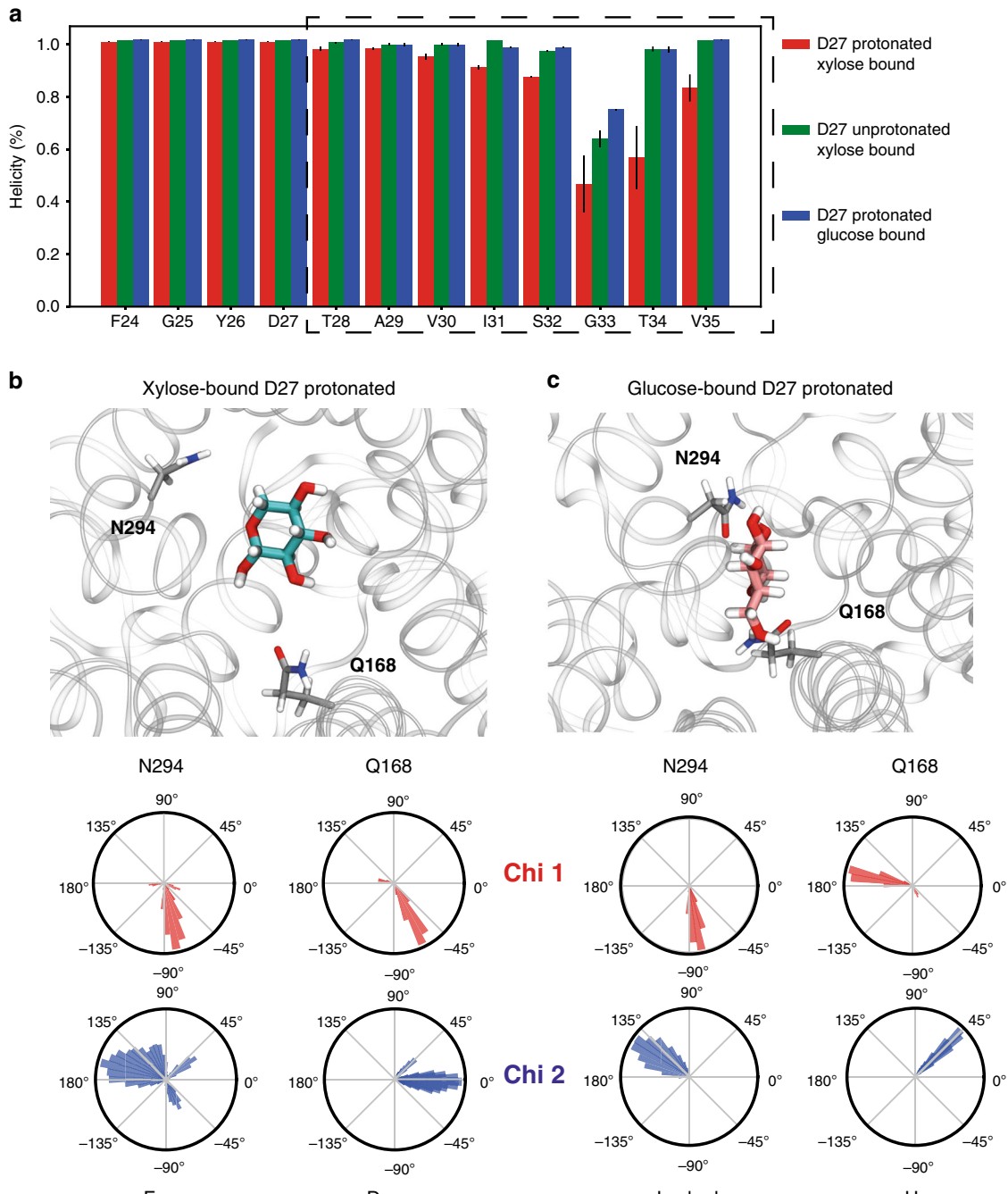

**Fig. 6 Helicity content of TM1 and dihedral angle changes at substrate-binding site. a** Decrease in helical content is more pronounced in the case of protonated D27 + xylose-bound case (red) than unprotonated D27 (green) or when bound to glucose (blue). Differences in the orientation of N294 and Q168 based on the Chi 1 and Chi 2 dihedral maps generated from the combined trajectory of the two replicates. Protein and ligand are rendered from an equilibrated snapshot from MD trajectory at t > 100 ns. In **b**, N294 is freely moving and residue Q168 is pointing down and away from xylose, whereas in **c**, residue N294 is locked and Q168 is pointing up and towards glucose.

suggests that the combined presence of xylose and a proton on D27 leads to an unstable state, hereby called "transition state," which allows the conformational transition underlying transport.

## Discussion
As a symporter, XylE binds and co-transports protons alongside its substrate xylose. The coupling between ligand binding and conformational changes is central to active transport but the molecular determinants leading to the conformational transition

are difficult to assess experimentally. Our work demonstrates the capability of HDX-MS to identify the structural signature of such coupling. Combined with predictions from MD simulations, we can decipher the molecular details underlying the interplay between substrate and proton binding.

The most striking result of this work shows that XylE variant D27N leads to a transition state only if xylose is already bound, highlighting an allosteric coupling between the substrate-binding pocket and the charge network. This effect is specific to xylose and shows that the protein can distinguish between substrate and

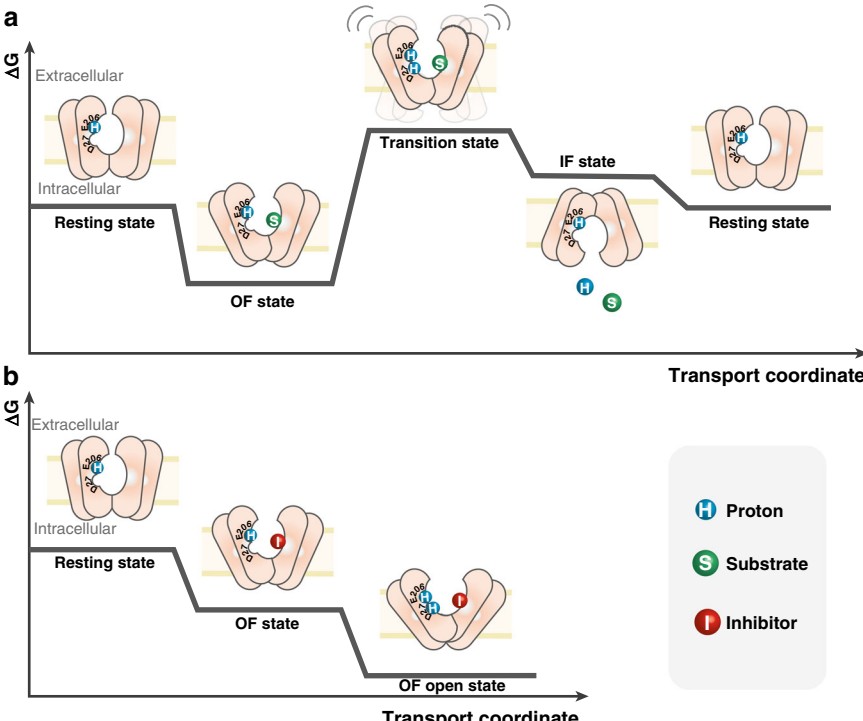

**Fig. 7 Ligand-dependent energy landscape of XylE. a** Proposed model of the XylE energy landscape in the presence of the substrate xylose. In the resting state, XylE is protonated at E206. Substrate binding stabilizes the OF state. Subsequent protonation of D27 leads to a dynamic "transition state" that allows the conformational transition towards the IF state. After substrate and proton release, the transporter switches back to the resting state. **b** Proposed model of energy landscape of XylE in the presence of inhibitor glucose. Inhibitor binding stabilizes the OF state. Subsequent protonation of D27 further stabilizes the OF state, effectively locking the transporter and preventing the conformational transition.

inhibitor. Furthermore, we observe that xylose or glucose binding is protonation independent and favours the OF conformation, in line with the OF ligand-bound structures obtained at basic conditions (pH 9.6)[17]. It is worth noting that the mutations we have used as proxies for protonation, while revealing important effects, have their limits. The mutation is permanent, whereas protonation is an equilibrium reaction that depends on solvent accessibility and local $pK_a$ values, which are likely to change during the conformational cycle[24]. Furthermore, D27N mutant is known to be functionally inactive, as demonstrated in cell-based uptake assays[23]. However, such mutants have already been used successfully to decipher the molecular mechanism of other proton-coupled transporters such as the MDR transporters AcrB, LmrP, PfMATE and MdfA[33–36], and identified key structural motifs during the transport cycle. Comparative HDX-MS experiments of protein harbouring protonation mimics appears to be a valuable method to study the molecular mechanism of proton-coupled transporters.

We propose the following transport cycle (Fig. 7): in its resting state, the WT transporter is protonated at residue E206 most or all of the time, in agreement with the high $pK_a$ values observed for E206 in both outward and inward conformations. Binding of xylose to the protonated transporter stabilizes the OF conformation and facilitates solvent accessibility to residue D27 (Figs. 7a and 5). The protonation of D27 when xylose is bound leads to a high-energy transition state, which initiates the conformational switch. This transition state is accessible only through allosteric coupling between D27 and the substrate-binding site, and such coupling is exquisitely sensitive to xylose binding. Under transport conditions (e.g., in the presence of a proton gradient), XylE can then switch toward the IF conformation and release substrate and proton in the cytosol. In contrast, binding of an inhibitor such as glucose further stabilizes the OF conformation, effectively trapping

the transporter in an energy well and preventing the conformational cycling required for transport (Fig. 7b). The identification of D27 as the driver of the conformational transition correlates with the known role of equivalent residues for other proton-coupled MFS transporters such as LacY (E325), LmrP (E327), MdfA (D34) and YajR (E320)[37]. This suggests a conserved mechanism of action among proton-coupled symporters of the same structural family.

We surmise that, along the resolution revolution, the development of tools and workflows capable of answering mechanistic questions at a molecular level is much needed and we demonstrate that HDX-MS coupled to MD simulations have a key role to play.

## Methods

**XylE expression and purification**. XylE was overexpressed in *E. coli* BL21-AI (DE3) (Invitrogen), which was transformed with the xyle gene in the presence or absence of the chosen mutations and cloned in the (30 μg/ml) kanamycin-resistant pET28-a plasmid (Novagen) modified with a C-terminal ten-histidine tag, grown in six baffled flasks each containing 1 L of Lysogeny Broth (LB) media at 37 °C 220 r.p.m. to an $OD_{600}$ of 0.8. Expression was induced with 1 mM isopropy-β-D-1-thiogalactopyranoside and 0.1% (w/v) L-arabinose, and growth continued until no increase of $OD_{600}$ is observed. The cells were collected by centrifugation, washed in 200 mL phosphate-buffered saline (PBS) buffer and centrifuged again for 20 min at 4200 r.p.m. in a Beckman JLA-16.250 rotor. The pellet was then resuspended in 50 mL PBS with 10 mM β-mercaptoethanol and 1 cOmplete protease inhibitor tablet and was frozen at −70 °C before purification. Cells were defrosted and incubated with 1.5 μL benzonase nuclease (ThermoFisher) for 10 min at room temperature before passed through constant cell disrupter at 25 kPsi, 4 °C. Then the ice-chilled membranes were isolated by ultracentrifugation for 30 min at 38,000 r.p.m. in a Beckman Ti45 rotor, 4 °C. Membrane pellets were solubilized for 2 h with mixing in solubilization buffer [50 mM sodium phosphate pH 7.4, 200 mM NaCl, 10% (v/v) glycerol, 20 mM imidazole, 10 mM β-mercaptoethanol and 2% *n*-dodecyl-β-D-maltoside (β-DDM, Anatrace), 0.1 mM phenylmethylsulfonyl fluoride (PMSF) and EDTA-free protease inhibitor tablet (Roche)] at 4 °C. Then the protein solution was isolated by centrifugation for another 30 min at 38,000 r.p.m. in a Beckman Ti70 rotor, to remove DDM insoluble material. The supernatant was

filtered using 0.45 μm filter and applied to a Ni-NTA column equilibrated in 96% size exclusion chromatography (SEC) purification buffer [50 mM sodium phosphate pH 7.4, 10% (v/v) glycerol, 2 mM β-mercaptoethanol, 0.05% β-DDM (Anatrace), 0.1 mM PMSF] and 4% elution buffer [50 mM sodium phosphate pH 7.4, 500 mM imidazole, 10% (v/v) glycerol, 10 mM β-mercaptoethanol, 0.1 mM PMSF and 0.05% β-DDM (Anatrace)]. The bound protein was washed with 50 mL 85% SEC purification buffer, 15% elution buffer and eluted with 2 mL of 100% elution buffer, which was collected for further SEC. The SEC purification was conducted with a Superdex 16/600 GL SEC column, which was equilibrated with SEC purification buffer. The elution fraction was collected and concentrated with a Vivaspin concentrator (100 kDa cutoff) (Supplementary Figs. S10 and S11). The samples were either flash frozen and kept at −70 °C until use or were used directly for HDX-MS experiments.

**Hydrogen-deuterium exchange mass spectrometry**. HDX-MS experiments were done using a Synapt G2-Si HDMS coupled to nanoACQUITY UPLC with HDX Automation technology (Waters Corporation, Manchester, UK)[11]. Membrane proteins in detergent micelles were prepared at a protein concentration around 30 μM using a 100 kDa cutoff Vivaspin concentrators. Before quenching in 100 μL ice-cold buffer Q (100 mM phosphate in formic acid pH 2.4), each 5 μL protein sample was incubated for 30 s, 5 min and 30 min in 95 μL deuterium-labelling buffer L (10 mM potassium phosphate in $D_2O$ pD 7.0).Then, the protein was digested with self-packed pepsin column at 20 °C. The reference controls were performed using the same protocol, with incubation with 95 μL equilibration buffer E (10 mM potassium phosphate in $H_2O$ pH 7.0) instead. The pepsin column was washed between injections using pepsin wash buffer (1.5 M Gu-HCl, 4% (v/v) MeOH, 0.8% (v/v) formic acid). A wash run using a saw-tooth gradient was done between each sample run to reduce peptide carry-over. Peptides were trapped for 3 min using an Acquity BEH C18 1.7 μM VANGUARD pre-column at a 200 μL/min flow rate in buffer A (0.1% formic acid in HPLC water pH 2.5) before eluted to an Acquity UPLC BEH C18 1.7 μM analytical column with a linear gradient buffer B (8–40% gradient of 0.1% formic acid in acetonitrile) at a flow rate of 40 μL/min. Then, peptides went through electrospray ionization progress in a positive ion mode using Synapt G2-Si mass spectrometer (Waters). Leucine Enkephalin was applied for mass accuracy correction and sodium iodide was used as calibration for the mass spectrometer. HDMS$^E$ data were collected by a 20–30 V trap collision energy ramp. All the isotope-labelling time points were performed in triplicates.

**HDX data evaluation and statistical analysis**. Acquired reference MS$^E$ data were analyzed by PLGS (ProteinLynx Global Server 2.5.1, Waters) to identify the peptic peptides, then all the HDMS$^E$ data including reference and deuterated samples were processed by DynamX v.3.0 (Waters) for deuterium uptake determination. Peptide filtration and analysis were performed as described before[11]. Woods plots were generated using Deuteros software[27,28].

**CD measurements**. CD thermal denaturation was performed in an Aviv Circular Dichroism Spectrophotometer, Model 410 (Biomedical, Inc., Lakewood, NJ, USA). All samples of XylE were measured at a protein concentration of 0.14–0.17 mg/ml and using a cell path length of 1 mm. The sample was heated at 5 °C intervals in SEC purification buffer (50 mM sodium phosphate, 10% (v/v) glycerol, 2 mM β-mercaptoethanol, and 0.05% β-DDM (Anatrace), 0.1 mM PMSF pH 7.4) from 25–95 °C. Each sample was scanned two times at a fixed wavelength of 222 nm in 1 nm wavelength steps with an averaging time of 1 s. The mean residue ellipticity ($[\theta]_{mre}$) at 222 nm was used for further analysis and is calculated using the following equation:

$$[\theta]_{mre} = \frac{MRW \times \theta_{obs}}{10 \times d \times c} \quad (1)$$

where $\theta_{obs}$ is the observed ellipticity in degrees, $d$ is the path length in cm, $c$ is the concentration in mg/ml. The mean residue weight (MRW) (~110 for most proteins) is calculated as the molecular mass divided by the number of amino acids − 1.

**Molecular dynamics: simulation setup**. MD simulations were initiated from either xylose (PDB ID: 4GBY) or glucose-bound (PDB ID: 4GBZ) state of XylE[17]. Protonation states of the titratable residues were assigned based on p$K_a$ calculations performed using PROPKA3.1 at pH 7[38]. Thereafter, XylE was embedded in a POPE lipid bilayer using the membrane replacement method in CHARMM-GUI[39]. System was solvated with TIP3P water molecules[40]. Thereafter, Na$^+$ and Cl$^-$ ions were added, and the system was neutralized with the ionic concentration set to 100 mM. The final system inclusive of the protein, lipids, water molecules and ions comprised ~100 K atoms.

Subsequently, the system was relaxed by minimizing it to a minimum for 5000 steps using conjugate-gradient algorithm and simulated for 5 ns at 310 K, with all the heavy atoms of the protein and the substrate restrained to their crystallographic positions with a force constant of $k = 5$ kcal/mol/Å$^2$. Finally, all the restrains were removed and the systems were simulated for 500 ns.

**MD simulation protocol**. The simulations were performed on with NAMD 2.13[41] employing CHARMM36 protein and lipid forcefields[42]. Simulations were performed in an NPT ensemble with periodic boundary conditions. Temperature was maintained at 310 K using Langevin dynamics with a damping constant of 0.5 ps$^{-1}$. Pressure was maintained at 1 atm using the Nosé–Hoover Langevin piston method[43]. The cutoff used for the short-range interactions were 12 Å with the switching applied at 10 Å. Long-range electrostatics was treated by the employing particle mesh Ewald (PME) algorithm[44]. Bonded, non-bonded and PME calculations were performed at 2, 2, and 4 fs intervals, respectively.

**Analysis: dynamical network analysis**. In XylE, coupling in the extracellular and intracellular gates can be understood in terms of the allosteric interactions of residues that efficiently move in a correlated manner. For this, dynamic network analysis was performed using the Network-View plugin[45] in VMD. In a network, all Cα carbons are defined as nodes connected by edges if they are within 4.5 Å of each other for at least 75% of the MD trajectory. Pearson's correlation was used to define the communities in the entire network corresponding to the set of residues that move in concert with each other (Supplementary Figs. S12 and S13).

**Reporting summary**. Further information on research design is available in the Nature Research Reporting Summary linked to this article.

## Data availability

Data supporting the findings of this paper are available from corresponding authors upon reasonable request. All the deuterium uptake plots of the experiments presented for XylE are available on figshare data repository using the following link: (https://figshare.com/s/52d498fe3b10c60b64a4). Spectrometry proteomics data have been deposited to the ProteomeXchange Consortium via the PRIDE partner repository with the dataset identifier PXD018145.

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

## Acknowledgements

We acknowledge funding from the Leverhulme Trust (RPG-2019-178) and the Wellcome Trust (109854/Z/15/Z) to A.P., the European Research Council, ERC Advanced grant 294342 and Wellcome Trust Investigator Award 214259/Z/18/Z to P.B., as well as thank King's College London for support for G.P. We also thank King's College London and China Scholarship Council for providing Ph.D. studentship for R.J. This research was supported by the National Institutes of Health under awards R01-GM123455 and P41-GM104601 from the National Institute of General Medical Sciences (to E.T.). S.P. acknowledges receiving support from the Beckman Institute Graduate Fellowship. We acknowledge the computing resources provided by Blue Waters of National Center for Supercomputing Applications (award ACI-1713784 to E.T.) and by Extreme Science and Engineering Discovery Environment (XSEDE award MCA06N060 to E.T.). C.M. benefits from a "Chargé de Recherches" fellowship of the FRS-FNRS.

## Author contributions

C.M. and A.P. designed the research. R.J. performed and analyzed HDX-MS experiments. R.J., H.F., G.P., N.H. and P.B. prepared samples and performed CD measurements. M.S., S.P. and E.T. performed and analysed MD simulations. A.L. provided support with HDX-MS analysis. C.M., R.J., M.S., S.P. and A.P. wrote the manuscript with input from all authors.

## Competing interests

The authors declare no competing interests.
