## [Peer Review File · Nature Communications]

REVIEWER COMMENTS

Reviewer #1 (Remarks to the Author):

- Good introduction of the HDX-MS and its role in structural biology as well as the structural understanding of XyleE but missing some more general background on XyleE. This is limited to one general statement
- State that peptides generated can be different between biological replicates but no mention that peptides can be different based on mutant constructs
- Figure 1 is much needed to understand all the combinations that will be studied in the paper
- What is the molar ratio of protein to substrate/inhibitor? Is binding saturated? What is the binding affinity of glucose to WT and is it different with the mutants? Only xylose binding affinity is discussed.
- In Figure 3 when you compare the role of substrate and inhibitor binding the differential HDX experiments all refer to WT thus they are showing both ligand binding as well as mutation changes. You already presented the mutation changes in Fig 2, so it would be best to set-up the differential in Fig 3 to only be +/- ligand such that the changes observed are only due to the ligand binding.
- The interpretation of Figure 4A needs to be reworked.
 - o "the increased uptake on both sides of the transporter indicate that there is an increase in local structural dynamics on the entire protein." An increase in dynamics across the entire protein would not yield differential HDX pattern
 - o "active state" Why is this considered active? No experiments or references are made to the actual activity of the protein. Needs to be more specific with the wording and/or explanation
- Figure 4C reference in text should be Figure 4E when discussing glucose binding
- Figure 5 is hard to read as is and needs to be better labeled and explained in the legend
 - o A-C: hard to see the difference in cyan vs yellow of the xylose and glucose. Change yellow?
 - o D-F legend says both xylose and glucose but graphs are all labeled xylose RMSD. I believe F should be glucose. Would be helpful to give title to all the subfigures or make clear that A+D+G, B+E+H, and C+F+I are all grouped together.
 - o G-I you should make the solvation spheres another color because red is used already
 - o J/K add label for yellow arrow in the legend
- Confusion of the pH of HDX experiments. The results section say pH7.4 but the method sections says D2O buffer pH6.6 and H2O buffer pH7.0, then the supplemental HDX info says pH7.0
- Discussion suggests that a highly conserved motif A on the cytoplasmic face could play a role in proton transfer. Was this not visible in the HDX experiments? Lack of sequence coverage?
- Figure comments
 - o Use of vs. instead of – to more accurately title the differential HDX experiment results
 - o Label all Ext and Int
 - o Figure reference need to be consistent. Capitalize or lower case all letters

Reviewer #2 (Remarks to the Author):

Summary:

In this manuscript, the authors use two complementary techniques, HDX-MS and MD simulation, to interrogate the effects of protonation and substrate binding on the transport cycle of the proton-coupled xylose transporter XyleE. The authors employed extensive HDX-MS experiments to reveal different patterns of protection/solvent accessibility across different conditions—namely, in the presence of xylose or glucose (an inhibitor) and/or mutations to two different titratable residues. First, the authors simply compare the WT, D27N, E206Q, and D27N/E206Q mutants to each other, and infer that protonation of D27N favors formation of the inward facing conformation because the differences in protection between D27N and D27N/E206Q were small, whereas the differences in protection levels between E206Q and the double mutant were large. Next, the authors add either xylose (the substrate)

or glucose (the inhibitor) to examine their effects on the four Xyle variants (WT and mutants). Adding xylose and glucose both stabilize the outward-facing conformation, regardless of the mutational background of the transporter. Finally, the combination of D27N and xylose compared to that of WT and xylose leads to a marked increase in deprotection throughout the transporter, suggesting that the combination of xylose and D27 protonation substantially alters the conformational landscape of Xyle. MD simulations started from the outward-occluded conformation bound to xylose, in the D27-protonated and D27-unprotonated conditions, or to glucose in the D27-protonated condition, indicate substantial increases in mobility of xylose in the the D27-protonated condition, in agreement with the findings from HDX-MS.

In general, this work uses an elegant combination of experimental and computational techniques to shed light on the conformational landscape of a transporter and is likely of broad interest to structural biologists trying to understand coupling between conformational changes and substrate binding. It will be of particular interest to those hoping to study transporters while avoiding use of the large probes and mutations typically required for FRET and DEER studies. My main comments are related to the clarity of the manuscript, including data presentation, and to the extent to which certain claims can be made given the existing data. I also think that the simulations are under-analyzed and could be mined carefully to extract atomic-level insights into the mechanisms by which glucose, xylose and D27 protonation affect the conformation of the transporter (e.g. its ability to adopt fully occluded or inward-facing states).

Major Comments:

- 1) Interpretation of HDX-MS experiments: the HDX-MS summary figures (really, color maps showing protection differences) seem to abstract away potentially useful information likely present in the raw data. The presentation gives the impression that the data obtained from HDX-MS is simply binary (protected or not protected relative to a reference). For example, are there any statistically significant differences in the degree of changes in deprotection or protection within certain regions of the protein across different conditions? Commenting on your choice of analysis method in more detail, and including some representative peptides' deuterium uptake plots or raw peptide spectra in the supplement, might aid readers' ability to interpret and understand your data (as suggested by the recent Nature Methods paper by Masson et al., 2019 on best practices in publishing HDX data).
- 2) An example of where more detailed information could aid the reader comes in Figure 2D: you conclude that the differences between D27N/E206Q and D27N are minor, but a region of increased protection on the intracellular side reveals significantly increased protection in the D27N/E206Q mutant. Could you comment on how you would interpret this?
- 3) Additionally, you show differential HX plots for many conditions using differences from WT or from other mutants, but I am curious also about the control condition mentioned: that of the G->W outward open-state stabilizing mutation. Can you infer anything about the relative frequency of the outward-open state under the WT condition and some of the other conditions (e.g. +xylose, D27N and D27N+xylose) using the G->W mutation as a reference (assuming that the mutation locks the transporter in the outward-open state)?
- 4) Could the decreased protection seen for the D27N+xylose - WT+xylose difference map result from increased cycling through the transport cycle, perhaps explained by a destabilization of the outward-facing state? I am uncomfortable with labeling the resulting D27N+xylose equilibrium an 'active state' since the term 'active state' has many different implications (e.g. for cell signaling, this typically indicates a state that enables binding of a downstream partner or catalysis of another reaction). Moreover, in reality, the transporter is not actually protonated at D27 throughout the transport cycle; instead, D27 likely switches protonation state, driving conformational changes.
- 5) Although the MD simulation results broadly appear to agree with the HX data, I have a few comments here related to their use and interpretation, which felt somewhat sparse.
 - a) First, I recommend avoiding any sentences that suggest that D27 must be protonated before xylose can bind, as this seems to contradict the fact that a crystal structure—and simulations of deprotonated D27—can stably bind xylose. I think it makes sense to say that D27 protonation triggers the

transition, but I don't know if the evidence you have is strong enough to claim an 'order of events.'

b) Along these same lines, if you have the resources, I would recommend including additional simulations of two key states: the D27-deprotonated and D27-protonated conditions without xylose (i.e. in the unliganded state), with an emphasis on the latter. In particular, I would expect that simply protonating D27 might perturb contacts and polar networks in the vicinity of the binding pocket, even if your HX-MS results indicate an overall reduction in deuterium uptake. Knowing the extent to which these changes affect the dynamics of residues in the extracellular gates might facilitate additional comparisons to the HDX-MS data. Perhaps you could then use the differences between the simulations with and without xylose in the D27-protonated condition to comment on the mechanism by which xylose has a particular effect on the transporter, triggering additional changes. In my experience, this sort of analysis often requires more than two simulations (three to five) per condition and a good amount of visual analysis of side-chain and backbone dynamics, which ties in to my next point. (The analysis can and should still be done even on just the two simulations you have.)

c) To complement the new simulations suggested above, I recommend that you expand your simulation analysis of the protein's conformational dynamics. Currently, the plots in Figure 5 only reveal differences in the stability of substrate. Please report which side chains in the vicinity undergo new rearrangements (e.g. by plotting certain dihedral angles or by calculating the frequency with which a side chain adopts a different rotamer). I'd be particularly interested in residues that link the ligand-binding pocket to the pocket of titratable residues (27, 133 and 206). For example, how does the helix break in helix 1 change across the different protonation conditions, if at all? How do changes in the position of helix 1 affect the relative orientations of, and the hydrogen bonding networks, formed by Thr28 and Gln175? Certain statistics, such as counts of waters in the binding pockets, etc. should be included in the supplement.

d) Related to the point above, Gln175 seems like it would be quite important for glucose affinity but not for xylose affinity, given the position of the sixth carbon on glucose. Again, I think the authors should be able to comment on the atomic-level basis for how xylose but not glucose has an effect beyond what they've shown here.

e) Finally, the Network Analysis results are quite difficult to interpret. Any two simulations might exhibit differences in residue correlations just based on stochastic fluctuations, and it is not clear how you took both simulations into account for each condition. Although I have not personally used these network analysis approaches, I believe some effort must be taken to quantify the statistical significance of differences between conditions, if the authors would really like to include these analyses. I don't find that they add to the manuscript.

6) Please comment in the Discussion on whether mutations of titratable acidic residues to Asn and Gln really recapitulate protonation observed during the transport cycle and what caveats might need to be appreciated to aid in the readers' interpretation of the data.

Minor Comments:

1) I found the statement that WT and D27N both bind xylose with a similar affinity to be confusing based on the sentence that follows it: "Biochemical studies and MD simulations have shown that protonation must occur before xylose binding, but it is not clear whether E206, D27 or both are protonated." If D27N has the same affinity as WT for xylose, then my quick interpretation would be that protonation of D27 is not required for xylose binding. Also, are you certain that these studies actually show temporal effects, i.e. that protonation must occur before binding?

2) Could the authors elaborate on the statement "Assuming that no major changes in the stability of transporters occur when introducing either the ligand or a mutation..." — in particular, for a non-biophysical audience, it might be worth explaining that if a protein increased substantially in stability, then it would take much longer for H-D exchange to come to equilibrium, and vice-versa?

3) Figure 1B: what is meant by a 'combination' of protein states? I'm having trouble calculating how you got 28 combinations based on the diagrams shown in Figure 1.

4) Is the reference 22 correctly referenced in the introduction?

5) The statement that "xylose or glucose binding induces a similar structural rearrangement towards the OF conformation regardless of the initial conformational state of the transporter" is a bit confusing. I don't necessarily disagree with the conclusion from the data, but I don't know what the 'initial'

conformation of any transporter is. I think the authors are trying to say that, regardless of whether a protonated variant (e.g. D27N) favors the inward conformation more strongly, xylose/glucose will still shift the conformational equilibrium to the outward facing state. In general, I think the authors could be a bit more careful with their choice of words when describing conformational equilibria.

6) Simulation figures: when showing protein structures, I recommend labeling all residues within the insets and showing a zoom-out of the protein to help with orientating the reader. Also, given the thickness of helices, I cannot really make out where the glucose is located.

Reviewer #3 (Remarks to the Author):

In the manuscript by Jia, Martens et al. the spatial dynamics of XylE variants in the presence and absence of xylose and glucose, respectively, are studied to elucidate the coupling between protonation and substrate transport. For this purpose HDX-MS was applied.

The data is well presented and easy to follow demonstrating a particular strength of HDX-MS data when compared to EPR for example. The interpretation of the HDX-MS and MD data comparing the presence of substrate xylose with inhibitor glucose when D27 is protonated (by use of D27N) is conclusive to me, showing that only in combination of the substrate and the protonation at position D27 a high spatial dynamics with IF and OF states is present resembling transporter motions. However, I am not convinced about the further interpretation of the data regarding the sequence of proton and substrate binding and I found some inconsistencies in the data I could not explain to me as detailed in the points below.

1. My main concern addresses your interpretation that the protonation at E206 is required for substrate binding (which due to its pKa is probably always the case) while a protonation of D27 afterwards is required for the conformational change. I think this interpretation originates from the observation that the presence of xylose alone is sufficient for stabilizing an OF conformation (indicating substrate binding in the absence of D27 protonation), while a 'turn-over' is only observed in the presence of the protonation. However, I disagree with this interpretation. First, I am missing data that prove that the protonation of E206 is required. For this binding assays or even HDX data with a disrupted E206 protonation site (e.g. E206A) would be required showing that this abolishes substrate binding. Second, it is known that secondary transporters exchange substrate without the coupling ion present (e.g. LacY). Hence, xylose should be able to bind in the absence of the coupling proton possibly triggering the stabilization of one conformation as observed. However, this does not prove that the protonation of D27 can still take place after the substrate has bound. To be honest I do not think that even a different setup of your experiments could address this question. You would need to perform transport assays with varying substrate availabilities and proton gradients to address this point as they probably have been performed before.

2. I am a bit puzzled about the low substrate concentration used in the experiments. If the affinity is 0.3 mM as mentioned, I would suggest the use of a tenfold excess ensuring saturation. You used 750 μ M of xylose or glucose so one can assume the presence of different populations. What was the reason for such low concentration and what is the affinity of XylE for glucose? Have you used different concentrations ensuring saturation? If not, the data in the presence of xylose and D27N could in principle represent states with different substrates (only proton or proton plus xylose bound).

3. When looking at the protonation-dependent conformational changes, D27N or E206Q show decreased deuterium accessibility at both sides compared to WT while the double mutant shows an increased uptake on the intracellular side and a decreased uptake on the extracellular side. Based on these data one would assume that both protonations are required. However, you next subtracted each single mutant from the double mutant coming to the conclusion that only the protonation of D27 is required for stabilizing the IF conformation. If so, why is this not visible when comparing the D27N with WT? Is this due to a different extent of conformational changes, which are not obvious from the

present representation? And if so, wouldn't it be helpful to show a gradient in color coding demonstrating the extent of conformational change?

4. You calculated the pKas of residues D27 and E206 coming to the conclusion that E206 is probably mostly protonated at your experimental conditions. However, you did the calculation only for one OF structure. What are the pKas for the other structures, particularly the inward facing?

Minor comments

5. Page 9: "The comparison between the protein in the presence and absence of xylose consistently shows that the presence of the substrate leads to a Δ HDX pattern typical of a transition towards an outward open conformation; an increase in deuterium uptake on the extracellular side coupled to a decrease in deuterium uptake on the intracellular side" The last extracellular should be intracellular.

6. The figure labeling is inconsistent with capital and small letters.

Reviewer #1 (Remarks to the Author):

- Good introduction of the HDX-MS and its role in structural biology as well as the structural understanding of XylE but missing some more general background on XylE. This is limited to one general statement.

We agree with the reviewer that the background information was succinct. We have now added information on XylE function and its relevance as a model for sugar import, with emphasis on how it compares to human GLUTs transporters.

The expanded section:

The symporter XylE of the ubiquitous MFS family is a bacterial homolog of human glucose transporters GLUTs 1-4 with an average sequence similarity of ~50% [1]. The xyle gene was first isolated in 1987 [2] and the expressed protein was shown to use the proton-motive force to specifically catalyse xylose translocation across the membrane of Escherichia coli [3]. The majority of bacterial sugar transporters relies on ion gradients to energize transport [4]. Mammalian GLUTs transporters in contrast are facilitators. This difference in sugar transport energetics between human and bacteria appears to arise from the scarcer availability of sugar for bacteria, compared to humans whose sugar levels in the blood are in the mM range [5]. Despite this difference, several residues are strictly conserved from XylE to GLUTs 1-4, critical either for substrate recognition or to enable structural rearrangements [3].

- State that peptides generated can be different between biological replicates but no mention that peptides can be different based on mutant constructs.

We observed that the pattern of peptides obtained is actually similar between the constructs (the peptides obtained for each XylE variant are reported on **Fig. S3**). However, the peptides containing the mutations were excluded from analysis. This is now specified in the main text.

- Figure 1 is much needed to understand all the combinations that will be studied in the paper.

We thank the reviewer for this comment. This figure has been through a lot of edits.

- What is the molar ratio of protein to substrate/inhibitor?

During D₂O labelling, 750µM of substrate/inhibitor are in presence of 1.5µM of XylE, thus the molar ratio is 500:1 (ligand: protein).

Is binding saturated?

Given the low affinity of XylE for its substrates (~0.3mM), we cannot guarantee that binding is saturated. The reason we do not work at saturating conditions is very trivial: for practical reasons, the protein and ligand are mixed together at high concentrations (30µM protein and 15mM xylose) before a 20x dilution in D₂O buffer for labelling.

However, because of the large excess of substrate, we can reasonably assume that the majority of the transporters are in a bound-form in the presence of substrate, compared to the apo conditions. The ΔHDX pattern observed upon xylose addition confirms this assumption, as it matches with the one observed for the benchmark condition XylE G58W

(the introduction of a G58W mutation blocks the OF conformation, as shown in a previous study [6]). We deduce that the Δ HDX observed in the presence of ligand reflects a shift in the conformational equilibrium towards the OF conformation caused by substrate binding.

Based on this comment and those from the other reviewers, we feel that a direct comparison with the benchmark condition is useful to provide. We have now added a **supplementary figure S1** (figure here below) that compares the uptake of representative peptides in the presence of the G58W mutation (benchmark), in the apo condition and with xylose. This figure shows that the Δ HDX pattern caused by the presence of ligand is similar to that caused by the G58W mutation, which confirms that xylose tips the conformational equilibrium toward the OF conformation upon binding.

What is the binding affinity of glucose to WT and is it different with the mutants? Only xylose binding affinity is discussed.

According to the literature, XylE D27N and XylE WT have a similar affinity for xylose, with a high K_d around 300 μ M (measured by ITC) XylE WT has a similarly weak affinity for glucose (~700 μ M) [1]. How the affinity for glucose is affected by the D27N mutation is not known, and we carried out an ITC measurement to address this question. Given the social

distancing constraints currently existing in the working place and the associated difficulties with purifying protein and performing experiments, we only carried an ITC experiment for the missing dataset. We observe that XylE D27N seems to have a higher affinity for glucose compared to xylose, which might explain why the inhibitor locks the transporter in the OF. This is however only one measurement, and a rather noisy one, so this value should be taken with a grain of salt. Furthermore, ITC binding experiments for transporters are challenging to perform. They have an overall low affinity for their substrates and the presence of detergent and co-purified lipids can influence the heat exchange reaction. As stated in a review by Boudker et al [7] “*Enthalpy changes associated with conformational transitions in the protein in response to ligand binding may offset those due to complex formation yielding final enthalpy changes that are low*”. Our HDX results show unambiguously that ligand binding (glucose or xylose) does lead to conformational changes. For this reason, we do not feel comfortable drawing any conclusion from these differences in binding affinity.

A transport assay would be ideal to assess the differences between the mutants and the ligands. We are currently developing an assay that measures active xylose transport by XylE reconstituted in proteoliposomes by coupling the import with an enzymatic reaction that produces a fluorescent substrate. This type of assay is notoriously difficult to set up (even more so with the current work limitations) and, we believe, goes beyond the scope of the present work.

- In Figure 3 when you compare the role of substrate and inhibitor binding the differential HDX experiments all refer to WT thus they are showing both ligand binding as well as mutation changes. You already presented the mutation changes in Fig 2, so it would be best to set-up the differential in Fig 3 to only be +/- ligand such that the changes observed are only due to the ligand binding.

We agree with the reviewer and have modified the figure. However, as the combination of the mutation/ protonation mimic and ligand binding could have led to a different Δ HDX profile, we think it is still useful to show the previous figure. We have now moved it to the supplementary figure (**Fig. S5**) and modified the text accordingly.

- The interpretation of Figure 4A needs to be reworked.

We have done our best to clarify our results, interpretation and terminology to meet the reviewers' requests regarding this section of the article. **Fig.4** has been modified to focus on the comparison between the effect of xylose versus glucose binding in combination with the D27N mutation.

o "the increased uptake on both sides of the transporter indicate that there is an increase in local structural dynamics on the entire protein." An increase in dynamics across the entire protein would not yield differential HDX pattern

We have now reworked this section to clarify our interpretation of that result. Our reasoning is the following:

We know from the Δ HDX experiment comparing the apo protein variants with the xylose bound variants that xylose addition tips the conformational equilibrium toward the OF conformation (**Fig. 3 and Fig. S5**), regardless of the presence of mutations. In this context, this Δ HDX experiment in the presence of xylose aims to detect *further* changes caused by the mutation, knowing that most transporters are already in the OF conformation. What we observe is that adding a mutation/protonation mimic on top of adding substrate leads to an increase of HDX on the entire protein (where exchange is observed), meaning that the H-bonds are destabilized. This pattern differs substantially from the same Δ HDX experiment carried out in the absence of substrate (**Fig. 2a** - d27n apo vs wt apo) where the mutation actually reduces exchange, increasing H-bond stability. Thus, this experiment identifies the coupling between substrate binding and D27 protonation, and its effect on local structural dynamics that translate into an overall increase in H/D.

We have now modified the main text accordingly:

This increased uptake on both sides of the transporter suggests that there is a decrease in H-bond stability on the entire protein, resulting from enhanced local structural dynamics and/or increase in solvent accessibility. (...) The shift toward this "transition state" clearly demonstrates that a specific allosteric coupling exists between the mutation/protonation of D27 and binding of the substrate xylose.

Taken together, these results suggest that once the substrate is loaded, the transporter is in the OF conformation and is primed for the protonation step that will allow the conformational transition. Our work and others show that this transition from OF to IF is initiated by proton binding at D27. In a physiological context, this transition would involve a short-lived transition state. We believe that our approach that mimics permanent protonation using a mutation allows to capture the signature of this transition state – that is: an increase in HDX on both

sides of the transporter. We surmise that this “signature” reflects the higher free Gibbs energy of such state, that translates into an increase in local structural dynamics – a hypothesis corroborated by the MD simulations. The MD simulations predict an increase in solvent accessibility and local unfolding events such as loss of helicity of helix 1 when a proton is present on D27 on the substrate bound structure.

o “active state” Why is this considered active? No experiments or references are made to the actual activity of the protein. Needs to be more specific with the wording and/or explanation.

We agree that “active state” was a misguided choice of words, pointed out by all reviewers (we meant to convey that the protein was more “wiggly”). We have now called it “transition state”, in line with enzyme terminology, since the transport cycle can be likened to an enzymatic reaction [8].

- Figure 4C reference in text should be Figure 4E when discussing glucose binding.

This has now been corrected. The figure is now split into main **Fig. 4** and supplementary **Fig. S6**.

- Figure 5 is hard to read as is and needs to be better labeled and explained in the legend
o A-C: hard to see the difference in cyan vs yellow of the xylose and glucose. Change yellow?

o D-F legend says both xylose and glucose but graphs are all labeled xylose RMSD. I believe F should be glucose. Would be helpful to give title to all the subfigures or make clear that A+D+G, B+E+H, and C+F+I are all grouped together.

o G-I you should make the solvation spheres another color because red is used already
o J/K add label for yellow arrow in the legend.

Figure 5 has now been modified, along with the entire molecular dynamics section, following all three reviewers’ suggestions. Here below is the new figure 5 which we believe is more readable.

a. Xylose bound - D27 unprotonated

b. Xylose bound - D27 protonated

c. Glucose bound - D27 protonated

Figure 5. Molecular Dynamics simulations reveal differences in ligand stability and water accessibility. (a, i-ii) MD simulations of xylose (shown in cyan and red sticks) bound XylE in the D27 unprotonated state, highlights that the bound substrate remains stable through the two independent 500 ns long MD simulations. (a, iii) water accessibility to the binding site is restricted. (b, i-ii) MD simulations of xylose-bound to protonated D27 shows an increase in substrate flexibility and in (b, iii) water accessibility to the binding site. (c, i-ii) Bound glucose molecule (shown in pink and red sticks) in D27 protonated XylE retains the

crystallographic pose with a similar pattern of substrate stability (**c, iii**) and solvation as the xylose bound unprotonated state. The stability was characterized by monitoring the RMSD of the xylose or glucose with respect to the crystal structure pose.

- Confusion of the pH of HDX experiments. The results section says pH7.4 but the method sections says D2O buffer pH6.6 and H2O buffer pH7.0, then the supplemental HDX info says pH7.0.

We thank the reviewer for pointing this out. All the HDX-MS experiments were carried out at pD 7.0 in the presence of D₂O and pH 7.0 for the undeuterated samples. This has now been corrected.

- Discussion suggests that a highly conserved motif A on the cytoplasmic face could play a role in proton transfer. Was this not visible in the HDX experiments? Lack of sequence coverage?

We have a good sequence coverage of the entire protein, including peptides encompassing motif-A like polar networks (residues D337-R341-E397-R404 and Q80-R84-E153). These regions display significant Δ HDX but we believe that changes in these regions are caused by shifts in the conformational equilibrium, as the intracellular side gets more or less exposed during the OF/IF transition.

Since the HDX experiments were performed at constant pH and not under pH gradient conditions (required for active transport) we do not expect to observe proton transfer. We hypothesize that the combination of D27N mutation and xylose binding at constant pH allows to observe a state that would be transient (the transition state) under transport/physiological conditions. In a previous study, we have shown that perturbing the motif A by introducing conservative mutations mimicking protonation (D337N, E153Q, and E397Q – fig here below from reference [6]) affects the conformational equilibrium by favouring the IF conformation – by opposition to the effect of substrate binding. This previous data combined with the data obtained in the current study allow us to present a complete transport cycle, that starts from the substrate and proton loading and ends with the disruption of the motif A and the ensuing conformational changes that lead to substrate and proton release.

- Figure comments

o Use of vs. instead of – to more accurately title the differential HDX experiment results

o Label all Ext and Int

o Figure reference need to be consistent. Capitalize or lower case all letters

We have modified the figures accordingly.

Reviewer #2 (Remarks to the Author):

Summary: In this manuscript, the authors use two complementary techniques, HDX-MS and MD simulation, to interrogate the effects of protonation and substrate binding on the transport cycle of the proton-coupled xylose transporter XylE. The authors employed extensive HDX-MS experiments to reveal different patterns of protection/solvent accessibility across different conditions—namely, in the presence of xylose or glucose (an inhibitor) and/or mutations to two different titratable residues. First, the authors simply compare the WT, D27N, E206Q, and D27N/E206Q mutants to each other, and infer that protonation of D27N favors formation of the inward facing conformation because the differences in protection between D27N and D27N/E206Q were small, whereas the differences in protection levels between E206Q and the double mutant were large. Next, the authors add either xylose (the substrate) or glucose (the inhibitor) to examine their effects on the four XylE variants (WT and mutants). Adding xylose and glucose both stabilize the outward-facing conformation, regardless of the mutational background of the transporter. Finally, the combination of D27N and xylose compared to that of WT and xylose leads to a marked increase in deprotection throughout the transporter, suggesting that the combination of xylose and D27 protonation substantially alters the conformational landscape of XylE. MD simulations started from the outward-occluded conformation bound to xylose, in the D27-protonated and D27-unprotonated conditions, or to glucose in the D27-protonated condition, indicate substantial increases in mobility of xylose in the D27-protonated condition, in agreement with the findings from HDX-MS.

In general, this work uses an elegant combination of experimental and computational techniques to shed light on the conformational landscape of a transporter and is likely of broad interest to structural biologists trying to understand coupling between conformational changes and substrate binding. It will be of particular interest to those hoping to study transporters while avoiding use of the large probes and mutations typically required for FRET and DEER studies. My main comments are related to the clarity of the manuscript, including data presentation, and to the extent to which certain claims can be made given the existing data. I also think that the simulations are under-analyzed and could be mined carefully to extract atomic-level insights into the mechanisms by which glucose, xylose and D27 protonation affect the conformation of the transporter (e.g. its ability to adopt fully occluded or inward-facing states).

Major Comments:

1) Interpretation of HDX-MS experiments: the HDX-MS summary figures (really, color maps showing protection differences) seem to abstract away potentially useful information likely present in the raw data. The presentation gives the impression that the data obtained from HDX-MS is simply binary (protected or not protected relative to a reference).

For example, are there any statistically significant differences in the degree of changes in deprotection or protection within certain regions of the protein across different conditions? Commenting on your choice of analysis method in more detail, and including some representative peptides' deuterium uptake plots or raw peptide spectra in the supplement, might aid readers' ability to interpret and understand your data (as suggested by the recent Nature Methods paper by Masson et al., 2019 on best practices in publishing HDX data).

We thank the reviewer for their comment. The binary plots presented show the regions that consistently displayed a significant ΔHDX over three or more biological replicates. We have performed statistical analysis of each ΔHDX -MS experiment using Deuterios and Deuterios 2.0 and only peptides showing a differ that is significant (confidence interval of 99%) were

kept [9, 10]. This is now clarified in the main text. We chose to present the data in a binary way because the sample to sample variability between biological replicates of membrane proteins is high compared to most soluble proteins – which makes biological replicates ever more crucial but also limits the quantitative analysis that can be performed. Our experience with HDX-MS on membrane proteins clearly indicates that regions with significant Δ HDX stay the same between biological replicates, but the absolute exchange values vary (see **Fig. S3** for woods plots of the biological replicates). We however agree with the reviewer that this representation obscures a lot of features that might be of interest. To address this shortcoming, we have included additional data both in the main manuscript and as supplementary data.

1. We have added heat maps of the relative fractional uptakes of all the conditions (**Fig. S2**), in addition to the woods plots (**Fig. S3**) and uptake plots that were already available supplementary data (link <https://figshare.com/s/52d498fe3b10c60b64a4>). These heat maps will allow the reader to see at a glance which regions exchange more, consistency between replicates and difference between conditions.
2. We have expanded the main figures to include additional data. We have added representative Woods plots on **Fig. 2** (effect of mutation) and **Fig. 4** (distinction between xylose and glucose) and representative peptides on **Fig. 3** (effect of ligand binding).

These peptides (1 on the IC side, 1 on the EC side for each Δ HDX experiment) were selected based on two criteria. First, they were identified as conformational reporters based on a benchmarking experiment performed previously that compared Xyle mutant G58W (locked in the OF) with WT [6, 11]. Second, those peptides – or fragments of it - had to be found in all the differential experiments presented.

We hope that these modifications will provide a clearer view of the link between the experimental data and the actual results.

2) An example of where more detailed information could aid the reader comes in Figure 2D: you conclude that the differences between D27N/E206Q and D27N are minor, but a region of increased protection on the intracellular side reveals significantly increased protection in the D27N/E206Q mutant. Could you comment on how you would interpret this?

The changes are significant but minor when compared to the changes observed between E206Q&D27N vs E206Q (effect of D27N). The extent of this difference is clearer when looking at the differential uptake data represented on a Woods plots, which have now been added on the figure **2**. Not only the number of peptides showing a significant Δ HDX but also the extent of the difference is larger (see new **Fig. 2** below – panels **c** and **d**). We agree with the reviewer that the increased protection caused by E206Q is significant and reproducible. We interpret this as a localized stabilization of the transporter upon E206 permanent protonation. Since the pK_a calculations based on the crystal structures predicts E206 to be protonated most of the time (pK_a 12), we hypothesize that the small changes reflect the difference between “protonated most of the time (no mutation)” and “protonated all of the time (E206Q)”.

Figure 2. Conformational change of XylE through mutation (protonation mimic). (a). Differential uptake pattern (Δ HDX) map comparing XylE E206Q or D27N mutants to the WT. (b). Δ HDX map between XylE E206Q&D27N and WT state. (c). Δ HDX map and woods plot between XylE E206Q&D27N and XylE E206Q. (d). Δ HDX map and Woods plot between XylE E206Q&D27N and XylE D27N. Each bar on Woods plots represents a single peptide with peptide length indicated by the bar length. Figures are projected onto topological and 3D protein structure (PDB: 4GBY) using PyMol. Blue and red regions indicate a negative (protected) and a positive (deprotected) deuterium uptake difference, respectively. Mutated residues are indicated by a star.

3) Additionally, you show differential HX plots for many conditions using differences from WT or from other mutants, but I am curious also about the control condition mentioned: that of the G->W outward open-state stabilizing mutation. Can you infer anything about the relative frequency of the outward-open state under the WT condition and some of the other conditions (e.g. +xylose, D27N and D27N+xylose) using the G->W mutation as a reference (assuming that the mutation locks the transporter in the outward-open state)?

This is a good idea. We could indeed use this condition to semi-quantify the extent of conformational cycling between the different mutants and ligand-bound states versus the locked OF state. This would require redoing the Δ HDX experiments, as the results obtained a couple of years ago on the G58W mutant cannot directly be compared with the datasets

presented in this work. This limitation comes not only from the sample to sample heterogeneity observed with membrane proteins, but also from instrument variability over time (how many runs have been through the LC and the pepsin columns, sensitivity and age of the MS, minor changes in the pH of the buffers, etc). We have however retrieved a dataset that compares the mutant G58W with WT apo and WT with xylose (performed the same week on the same WT sample). Peptides representative of regions identified as conformational reporters are presented on the figure here below. By looking at the data, it appears Xyle bound to xylose is not protected/deprotected to the same extent as the mutant, implying that the conformational equilibrium is not 100% skewed towards the OF state. This is in line with the comments from reviewer 1 and 3 suggesting that we are not at saturating concentration of substrate.

We thank the reviewer for this suggestion and we will implement this strategy for quantification in our upcoming work. The current restrictions on lab access however have prevented us from doing these experiments for this study but we believe that it does not impact our findings.

Figure. S1. Deuterium uptake plots of peptides identified as conformational reporters for Xyle [6]. The regions selected as conformational reporters are based on the Δ HDX between WT apo and G58W apo. Their location on the structure is shown on the IF conformation (PDB: 4JA3) with the extracellular reporters in blue and the intracellular reporters in orange.

4) Could the decreased protection seen for the D27N+xylose – WT+xylose difference map result from increased cycling through the transport cycle, perhaps explained by a destabilization of the outward-facing state?

That is a likely explanation that also fits with our simulations results (see answer to point 6c below). We surmise that Δ HDX measurements capture the dynamic signature of a state allowing for the conformational switch required for transport. Either increased conformational sampling or global destabilization (increased free energy) are compatible with that HDX behaviour. The difference between these two notions might be more about semantics.

I am uncomfortable with labeling the resulting D27N+xylose equilibrium an ‘active state’ since the term ‘active state’ has many different implications (e.g. for cell signaling, this typically indicates a state that enables binding of a downstream partner or catalysis of another reaction). Moreover, in reality, the transporter is not actually protonated at D27 throughout the transport cycle; instead, D27 likely switches protonation state, driving conformational changes.

We agree with the reviewer, along with the other reviewers, that “active state” was a poor choice of words and we are now labelling this putative state “transition state”. We also agree that D27 protonation is transient during the transport cycle. We hypothesize that using a mutant allowed us to trap an otherwise transient state, and observe an HDX signature exclusive to this state (see answer to reviewer 1, page6).

5) Although the MD simulation results broadly appear to agree with the HX data, I have a few comments here related to their use and interpretation, which felt somewhat sparse.

We have now expanded the MD simulation results section.

a) First, I recommend avoiding any sentences that suggest that D27 must be protonated before xylose can bind, as this seems to contradict the fact that a crystal structure—and simulations of deprotonated D27—can stably bind xylose. I think it makes sense to say that D27 protonation triggers the transition, but I don’t know if the evidence you have is strong enough to claim an ‘order of events.’

We agree with the reviewer that deriving a sequence of events from HDX measurements and MD predictions is a bit of stretch and have now shifted the focus of the manuscript toward the coupling between xylose binding and protonation of D27, and the specificity of such coupling. We also agree it is unlikely that D27 is protonated before substrate binding, and have removed sentences implying such thing. In the discussion section however, we examine our results in the context of previous studies. Two studies hypothesize that the transporter must be protonated before substrate can bind and propose D27 as the likely candidate [8, 12]. We try to reconcile their findings and ours by suggesting that XyleE can be protonated prior to xylose binding but not at D27. We propose that this initial protonation occurs at E206.

b) Along these same lines, if you have the resources, I would recommend including additional simulations of two key states: the D27-deprotonated and D27-protonated conditions without xylose (i.e. in the unliganded state), with an emphasis on the latter. In particular, I would expect that simply protonating D27 might perturb contacts and polar networks in the vicinity of the binding pocket, even if your HX-MS results indicate an overall reduction in deuterium uptake. Knowing the extent to which these changes affect the dynamics of residues in the extracellular gates might facilitate additional comparisons to the

HDX-MS data. Perhaps you could then use the differences between the simulations with and without xylose in the D27-protonated condition to comment on the mechanism by which xylose has a particular effect on the transporter, triggering additional changes. In my experience, this sort of analysis often requires more than two simulations (three to five) per condition and a good amount of visual analysis of side-chain and backbone dynamics, which ties in to my next point. (The analysis can and should still be done even on just the two simulations you have.)

As per the reviewer's suggestion we performed two additional set of MD simulations of XylE in apo-state, modulating the protonation state of the residue D27. The results are presented as projected XylE conformations onto a 2D space spanned by the distances between the intra and the extracellular gates (see figure below, also in supplementary **Fig. S12**). We observe that in the absence of Xylose, the protein has similar dynamics irrespective of the protonation state of the residue. As can be seen in the plot of the intra/extra cellular gate distances, simulations started from the apo state sample similar conformational landscape. However, presence of Xylose and the protonation of the residue D27 yields conformational heterogeneity that we believe triggers the transition to the IF state, evidenced by a closing of the extracellular gates.

c) To complement the new simulations suggested above, I recommend that you expand your simulation analysis of the protein's conformational dynamics. Currently, the plots in Figure 5 only reveal differences in the stability of substrate. Please report which side chains in the vicinity undergo new rearrangements (e.g. by plotting certain dihedral angles or by calculating the frequency with which a side chain adopts a different rotamer). I'd be particularly interested in residues that link the ligand-binding pocket to the pocket of titratable residues (27, 133 and 206). For example, how does the helix break in helix 1 change across

the different protonation conditions, if at all? How do changes in the position of helix 1 affect the relative orientations of, and the hydrogen bonding networks, formed by Thr28 and Gln175? Certain statistics, such as counts of waters in the binding pockets, etc. should be included in the supplement.

We agree and thank the reviewer for the request for the simulation analysis of the conformational dynamics of the protein. We performed two sets of analysis, focusing on both the global and local conformational dynamics. On the global scale, we projected XylE conformations onto a 2D space spanned by the distances between the intra and the extracellular gates (see gate analysis figure hereinabove). Briefly this analysis shows that in the absence of xylose the conformational dynamics of the protein irrespective of the protonation state of D27 is nearly identical. In contrast, in the presence of xylose, the protonation of D27 triggers the transition to the IF state, as evidenced by the sampling in the region of reduced intra and increased extra cellular gate distances. As per the request of reviewer, we explored the changes in the secondary structure feature of helix-1 upon changes in the protonation state. We observe a decrease in helical content in the presence of xylose or glucose. This decrease is significantly increase in the xylose-bound/D27 protonated case (figure here below, **Fig. 6a** in the manuscript). Henceforth, we analysed the dynamics and the stability of the salt-bridge network and the subsequent effects on the solvent exposure of the substrate binding site. Briefly, the perturbation of the salt bridge network upon the protonation of D27 results in the neighbouring residue T28 pointing to the substrate binding site (Figure here below, also in supplementary **Fig. S8**). As a result, there is a decrease in helicity of Helix 1, increased solvent access to the substrate and the protein binding site and subsequent destabilization of Xylose. This is now included in the main figures **5** and **6** in the manuscript. As an evidence for our hypothesis, we have added analysis pertaining to the water accessibility of the protein lumen as well as the conformational sampling of the residue D27, along with the key residues in the ligand binding site, in the supplementary data (**Fig. S13**).

Figure 6. Helicity content and dihedral angle changes. (a) Decrease in helical content is more pronounced in D27 protonated xylose bound case (red) than unprotonated D27 (green) or glucose bound (blue).

d) Related to the point above, Gln175 seems like it would be quite important for glucose affinity but not for xylose affinity, given the position of the sixth carbon on glucose. Again, I think the authors should be able to comment on the atomic-level basis for how xylose but not glucose has an effect beyond what they've shown here.

We have analysed all the dihedral angles changes of the residues important for binding in the xylose bound/D27 protonated cases, xylose-bound/D27 unprotonated cases and glucose-bound D27 protonated case (see figure below, also in supplementary **Fig. S9** and **Fig. 6b-c**). Surprisingly, we do not observe significant difference in the orientation of Gln175 in the presence of glucose compared to xylose. Altogether, the orientation of the residues is relatively similar in all three cases with two notable exceptions: residues Asn 294 and Gln168. These residues are not involved in xylose stabilization anymore when D27 is protonated.

Figure. S9. Dihedral angles of seven XyleE residues involved in substrate binding. Residues showing a different orientation in the xylose-bound D27 protonated case are highlighted in red.

e) Finally, the Network Analysis results are quite difficult to interpret. Any two simulations might exhibit differences in residue correlations just based on stochastic fluctuations, and it is not clear how you took both simulations into account for each condition. Although I have not personally used these network analysis approaches, I believe some effort must be taken to quantify the statistical significance of differences between conditions, if the authors would really like to include these analyses. I don't find that they add to the manuscript.

We agree with the reviewer that the network analysis does not add much, especially now that detailed analyses of the conformational space, the dihedral angles and solvent accessibility calculations have been included in the manuscript.

6) Please comment in the Discussion on whether mutations of titratable acidic residues to Asn and Gln really recapitulate protonation observed during the transport cycle and what caveats might need to be appreciated to aid in the readers' interpretation of the data.

We thank the reviewer for this suggestion and have now expanded the discussion. We insist on the fact that protonation is an equilibrium reaction, while mutation is a permanent modification. The modified text is the following:

It is worth noting that the mutations we have used as proxies for protonation, while revealing important effects, have their limits. The mutation is permanent while protonation is an equilibrium reaction that depends on solvent accessibility and local pKa values, which are likely to change during the conformational cycle[12]. However, such mutants have already been used successfully to decipher the molecular mechanism of other proton-coupled transporters such as the MDR transporters AcrB, LmrP, pfMATE and MdfA [13-16], and identified key structural motifs during the transport cycle. Comparative HDX-MS experiments

of protein harbouring protonation mimics appears to be a valuable method to study the molecular mechanism of proton-coupled transporters.

Minor Comments:

1) I found the statement that WT and D27N both bind xylose with a similar affinity to be confusing based on the sentence that follows it: “Biochemical studies and MD simulations have shown that protonation must occur before xylose binding, but it is not clear whether E206, D27 or both are protonated.” If D27N has the same affinity as WT for xylose, then my quick interpretation would be that protonation of D27 is not required for xylose binding. Also, are you certain that these studies actually show temporal effects, i.e. that protonation must occur before binding?

- This sentence was a rather crude summary of previous work on XylE and has now been removed. We refer to a study that uses SSM based electrophysiology to measure K_m as a function of pH [12]. The authors observe a decrease in affinity (increase in K_m) coupled to pH increase and deduce that H^+ binding occurs before xylose binding. Another study suggesting a similar order of events uses an analogy between LacY and XylE. Residue 325 of LacY is homologous to residue 27 of XylE, and it has been shown that E325 must be protonated before LacY substrate binding [8]. Altogether, these conclusions from previous work are rather hypothetical. We have now toned down all the aspects of temporal sequence of events in the manuscript, also because this scepticism was shared by all the reviewers.

2) Could the authors elaborate on the statement “Assuming that no major changes in the stability of transporters occur when introducing either the ligand or a mutation...” — in particular, for a non-biophysical audience, it might be worth explaining that if a protein increased substantially in stability, then it would take much longer for H-D exchange to come to equilibrium, and vice-versa?

We have now expanded that statement. H/D exchange kinetics depend directly on H-bond stability and a direct correlation exist between global stability and local dynamics [17], which is one of the reasons HDX has been historically used for protein folding studies.

3) Figure 1B: what is meant by a ‘combination’ of protein states? I’m having trouble calculating how you got 28 combinations based on the diagrams shown in Figure 1.

We have amended the figure, and added a table in the supplementary data that summarizes all the individual conditions tested. We have actually made 32 Δ HDX experiments, including measurements with glucose.

4) Is the reference 22 correctly referenced in the introduction?

We thank the reviewer for spotting this mistake, this reference has now been modified.

5) The statement that “xylose or glucose binding induces a similar structural rearrangement towards the OF conformation regardless of the initial conformational state of the transporter” is a bit confusing. I don’t necessarily disagree with the conclusion from the data, but I don’t know what the ‘initial’ conformation of any transporter is. I think the authors are trying to say that, regardless of whether a protonated variant (e.g. D27N) favors the inward conformation more strongly, xylose/glucose will still shift the conformational equilibrium to the outward

facing state. In general, I think the authors could be a bit more careful with their choice of words when describing conformational equilibria.

We thank the reviewer for their comment and have done our best to improve our writing. Regarding this specific statement we have now changed it:

We observe a systematic shift of the conformational equilibrium toward the OF state in the presence of either xylose or glucose. This transition is observed for all XylE variants, suggesting that substrate binding favors the OF conformation regardless of the prior protonation state of D27 or E206, and of the apo conformational ensemble of the transporter. These results are in line with the observed OF states of the ligand-bound structures captured by X-ray crystallography.

6) Simulation figures: when showing protein structures, I recommend labeling all residues within the insets and showing a zoom-out of the protein to help with orientating the reader. Also, given the thickness of helices, I cannot really make out where the glucose is located.

We have now changed the simulation figures and hope that the readability of the new figures has improved.

Reviewer #3 (Remarks to the Author):

In the manuscript by Jia, Martens et al. the spatial dynamics of XylE variants in the presence and absence of xylose and glucose, respectively, are studied to elucidate the coupling between protonation and substrate transport. For this purpose HDX-MS was applied. **The data is well presented and easy to follow demonstrating a particular strength of HDX-MS data when compared to EPR for example. The interpretation of the HDX-MS and MD data comparing the presence of substrate xylose with inhibitor glucose when D27 is protonated (by use of D27N) is conclusive to me, showing that only in combination of the substrate and the protonation at position D27 a high spatial dynamics with IF and OF states is present resembling transporter motions.** However, I am not convinced about the further interpretation of the data regarding the sequence of proton and substrate binding and I found some inconsistencies in the data I could not explain to me as detailed in the points below.

1. My main concern addresses your interpretation that the protonation at E206 is required for substrate binding (which due to its pKa is probably always the case) while a protonation of D27 afterwards is required for the conformational change. I think this interpretation originates from the observation that the presence of xylose alone is sufficient for stabilizing an OF conformation (indicating substrate binding in the absence of D27 protonation), while a 'turn-over' is only observed in the presence of the protonation. However, I disagree with this interpretation. **First, I am missing data that prove that the protonation of E206 is required.** For this, binding assays or even HDX data with a disrupted E206 protonation site (e.g. E206A) would be required showing that this abolishes substrate binding. Second, it is known that secondary transporters exchange substrate without the coupling ion present (e.g. LacY). Hence, xylose should be able to bind in the absence of the coupling proton possibly triggering the stabilization of one conformation as observed. However, this does not prove that the protonation of D27 can still take place after the substrate has bound. To be honest I do not think that even a different setup of your experiments could address this question. You would need to perform transport assays with varying substrate availabilities and proton gradients to address this point as they probably have been performed before.

We thank the reviewer for their comment, which is in line with the other reviewer's comments (see our answer to reviewer 2, page 14, who raised similar concerns). The sequence of events that we suggested is based on a combination of the work presented here and previous studies performed on XylE. We believe that the Discussion section is suitable for some speculation but have now made clearer what our work demonstrates and what is more hypothetical. Specifically, we have now toned down the temporal aspect of the transport cycle and focus on the specificity of xylose vs glucose coupling with protonation. The figure of the transport cycle has been modified to reflect that change (see below, **Fig. 7** in the manuscript).

Figure 7. Ligand-dependent energy landscape of XylE. (a) Proposed model of the XylE energy landscape in the presence of the substrate xylose. In the resting state, XylE is protonated at E206. Substrate binding stabilizes the OF state. Subsequent protonation of D27 leads to a dynamic “transition state” that allows the conformational transition towards the IF state. After substrate and proton release, the transporter switches back to resting state (b). Proposed model of energy landscape of XylE in the presence of inhibitor glucose. Inhibitor binding stabilizes OF state. Subsequent protonation of D27 further stabilizes the OF state, effectively locking the transporter and preventing the conformational transition.

2. I am a bit puzzled about the low substrate concentration used in the experiments. If the affinity is 0.3 mM as mentioned, I would suggest the use of a tenfold excess ensuring saturation. You used 750 μM of xylose or glucose so one can assume the presence of different populations. What was the reason for such low concentration and what is the affinity of XylE for glucose? Have you used different concentrations ensuring saturation? If not, the data in the presence of xylose and D27N could in principle represent states with different substrates (only proton or proton plus xylose bound).

The reason for using such low concentration was very trivial: since the labelling reaction dilutes the sample 20x, we started from a 15mM xylose solution to end up with 0.75 μ M under labelling conditions. We agree that this does not ensure saturation but think that it does not affect our interpretation of Δ HDX experiments, because of the observed shift in the conformational equilibrium that matches with our benchmark condition (see answer to reviewer 1 who raised similar concern - please find our answer on page 2.). We agree with the reviewer that the presence of different populations is likely but since Δ HDX experiments observe shifts in the conformational equilibrium, saturation is not an absolute requirement to be able to detect such population changes.

3. When looking at the protonation-dependent conformational changes, D27N or E206Q show decreased deuterium accessibility at both sides compared to WT while the double mutant shows an increased uptake on the intracellular side and a decreased uptake on the extracellular side. Based on these data one would assume that both protonations are required. However, you next subtracted each single mutant from the double mutant coming to the conclusion that only the protonation of D27 is required for stabilizing the IF conformation. If so, why is this not visible when comparing the D27N with WT? Is this due to a different extent of conformational changes, which are not obvious from the present representation?

We would like to clarify that we believe our results show that both residues have to be protonated to stabilize the IF conformation but that only D27 protonation on top of E206 protonation tips the conformational equilibrium towards the IF conformation. It is quite clear from our data that D27N does not lead to IF conformation, neither does E206Q. We have now changed the text accordingly: *taken together, these results suggest that D27 protonation is the main driver of the conformational transition to IF state, as long as a proton is already present on E206.*

We however agree with the reviewer that this result is surprising. Since pK_a calculations predict that E206 is protonated most of the time, one would assume that the D27N mutant would have a similar effect as D27N:E206Q. As pointed out by reviewer 2, mutations aiming at mimicking protonation have their limits. We surmise that each individual mutation might perturb the pK_a of neighbouring residues in a way that is not representative of what protonation does, and the effect of the double mutant is not the sum of the effect of each single mutant. The experiments comparing the double versus the simple mutants are however very clear at highlighting the specific conformational role of D27, which is the result we build upon.

And if so, wouldn't it be helpful to show a gradient in color coding demonstrating the extent of conformational change?

Showing the extent of HDX by using a color gradient is a good idea and is actually common practice in the HDX field. We have chosen not to do it because of the sample to sample variability in absolute uptake we routinely observe when working with membrane proteins (see answer to reviewer 2, page 10 and **Fig. S3** reporting all the Δ HDX of individual experiments).

4. You calculated the pK_a s of residues D27 and E206 coming to the conclusion that E206 is probably mostly protonated at your experimental conditions. However, you did the

calculation only for one OF structure. What are the pKas for the other structures, particularly the inward facing?

That's a fair point. Xyle has been crystallized in two inward facing conformations namely IF-occluded (PDB:4JA3) and IF-open (PDB:4JA4). In the IF-occluded the pKas of ASP27 and E206 are 3.64 and 10.99 respectively. On the other hand, in the IF-open conformation pKas of ASP27 and E206 are 3.64 and 11.11 respectively. Please see the figure here below, now included in the supplementary **Fig. S7**.

Minor comments

5. Page 9: "The comparison between the protein in the presence and absence of xylose consistently shows that the presence of the substrate leads to a Δ HDX pattern typical of a transition towards an outward open conformation; an increase in deuterium uptake on the extracellular side coupled to a decrease in deuterium uptake on the intracellular side" The last extracellular should be intracellular.

This has now been modified.

6. The figure labeling is inconsistent with capital and small letters.

This has now been modified.

1. Sun, L., et al., *Crystal structure of a bacterial homologue of glucose transporters GLUT1-4*. Nature, 2012. **490**(7420): p. 361-6.

2. Davis, E.O. and P.J. Henderson, *The cloning and DNA sequence of the gene xylE for xylose-proton symport in Escherichia coli K12*. J Biol Chem, 1987. **262**(29): p. 13928-32.
3. Quistgaard, E.M., et al., *Structural basis for substrate transport in the GLUT-homology family of monosaccharide transporters*. Nat Struct Mol Biol, 2013. **20**(6): p. 766-8.
4. Saier, M.H., Jr., *Families of transmembrane sugar transport proteins*. Mol Microbiol, 2000. **35**(4): p. 699-710.
5. Henderson, P.J., *Proton-linked sugar transport systems in bacteria*. J Bioenerg Biomembr, 1990. **22**(4): p. 525-69.
6. Martens, C., et al., *Direct protein-lipid interactions shape the conformational landscape of secondary transporters*. Nat Commun, 2018. **9**(1): p. 4151.
7. Boudker, O. and S. Oh, *Isothermal titration calorimetry of ion-coupled membrane transporters*. Methods, 2015. **76**: p. 171-182.
8. Madej, M.G., et al., *Functional architecture of MFS D-glucose transporters*. Proc Natl Acad Sci U S A, 2014. **111**(7): p. E719-27.
9. Lau, A.M.C., et al., *Deuterios: software for rapid analysis and visualization of data from differential hydrogen deuterium exchange-mass spectrometry*. Bioinformatics, 2019. **35**(17): p. 3171-3173.
10. Lau, A.M., et al., *Deuterios 2.0: Peptide-level significance testing of data from hydrogen deuterium exchange mass spectrometry*. Bioinformatics, 2020.
11. Martens, C., et al., *Integrating hydrogen-deuterium exchange mass spectrometry with molecular dynamics simulations to probe lipid-modulated conformational changes in membrane proteins*. Nat Protoc, 2019. **14**(11): p. 3183-3204.
12. Bazzone, A., et al., *pH Regulation of Electrogenic Sugar/H⁺ Symport in MFS Sugar Permeases*. PLoS One, 2016. **11**(5): p. e0156392.
13. Eicher, T., et al., *Coupling of remote alternating-access transport mechanisms for protons and substrates in the multidrug efflux pump AcrB*. Elife, 2014. **3**.
14. Masureel, M., et al., *Protonation drives the conformational switch in the multidrug transporter LmrP*. Nat Chem Biol, 2014. **10**(2): p. 149-55.
15. Claxton, D.P., et al., *Sodium and proton coupling in the conformational cycle of a MATE antiporter from Vibrio cholerae*. Proc Natl Acad Sci U S A, 2018. **115**(27): p. E6182-E6190.
16. Fluman, N., et al., *Dissection of mechanistic principles of a secondary multidrug efflux protein*. Mol Cell, 2012. **47**(5): p. 777-87.
17. Englander, S.W. and N.R. Kallenbach, *Hydrogen exchange and structural dynamics of proteins and nucleic acids*. Q Rev Biophys, 1983. **16**(4): p. 521-655.
18. Li, H., A.D. Robertson, and J.H. Jensen, *Very fast empirical prediction and rationalization of protein pKa values*. Proteins, 2005. **61**(4): p. 704-21.

REVIEWERS' COMMENTS

Reviewer #1 (Remarks to the Author):

The authors have addressed all of my concerns.

Reviewer #2 (Remarks to the Author):

Review for Nature Comms. 251578:

The revised manuscript by Jia et al. tells a clearer story than the original manuscript and presents additional representations of the HDX-MS data as well as additional analysis of the MD simulation data that together help to strengthen the paper.

The two major messages of the current story, as I interpret them, are that (1) protonation of D27 and xylose binding favor transitions out of the outward-facing state, as suggested by the HDX-MS data presented, particularly in Fig. 4., and (2) that the transporter can discriminate at the atomic level between xylose and glucose based on the fact that, in MD simulations, only the combination of D27 protonation and xylose binding significantly destabilize residues in the ligand-binding pocket.

I have only a few additional comments:

1. The authors propose to call the conformational ensemble favored by the combination of D27N and xylose, identified through the HDX-MS experiments, the "transition state." While I agree that this term is more fitting than their previous term ("active state"), the subsequent HDX-MS signal does not solely arise from the chemical transition state that represents the energetic barrier between two states. Rather, the HDX-MS experiments carried out under different conditions represent ensembles, or combinations of states weighted by their populations. I would therefore prefer if the authors used phrases like "[D27N+xylose compared to D27N alone] favors transition-competent conformations, where transition refers to the conformational change that allows the transporter to move between the outward-facing and inward-facing states."

2. Apologies if the authors included this and I missed it, but is the D27N transporter a physiologically functional transporter? Have the authors or others measured its function by liposome uptake assays or survival assays? If this data exists, I might recommend including mention of it in the Discussion, where the authors discuss such mutants.

3. The new simulation data, focused on differences in residue positions within the substrate-binding pocket, is exciting to see. I have a couple of questions and follow-up points:

(a) In Fig. 5, it is a bit difficult to tell whether xylose and glucose start out in the same pose and then xylose reorients when D27 is protonated, but glucose does not, or if xylose-bound and glucose-bound structures show different poses of the bound substrates. I imagine that the former is the case, but please clarify this somewhere in the text (or simply show the 'first' and 'last' frames instead of the entire ensemble of substrate positions in Fig. 5 (a-c, part i)--if the snapshots shown in Fig. 6 are in fact representative of those states, then please clarify this as well.

(b) Fig. 6a and Fig. S8a might both suffer from the same labeling confusion: in the former, the blue bars in Fig. 6a refer to a "D27 unprotonated, glucose-bound" condition, but I believe that this simulation has D27 protonated, for a comparison to the d27 protonated, xylose-bound condition? If not, then this label is inconsistent the labels in Fig. 5c. The same labeling issue might be happening in Fig. S8a.

(c) When showing the plots to demonstrate differences in dihedral conditions, please clarify whether those plots reflect data pooled across each of the two simulation replicates and indicate whether the simulations' behavior is consistent across the two replicates for a given condition.

(d) Finally, in the text, please clarify whether the changes in helix 1 conformation, and in the rotameric states of Q168 and N294, observed in simulation resemble these regions' conformations in

IF crystal structures? Or are these regions' conformations observed in simulation representative of a different conformational state?

Reviewer #3 (Remarks to the Author):

All my comments were satisfactorily taken into account.

Reviewer #1 (Remarks to the Author):

The authors have addressed all of my concerns.

Reviewer #2 (Remarks to the Author):

Review for Nature Comms. 251578:

The revised manuscript by Jia et al. tells a clearer story than the original manuscript and presents additional representations of the HDX-MS data as well as additional analysis of the MD simulation data that together help to strengthen the paper.

The two major messages of the current story, as I interpret them, are that (1) protonation of D27 and xylose binding favor transitions out of the outward-facing state, as suggested by the HDX-MS data presented, particularly in Fig. 4., and (2) that the transporter can discriminate at the atomic level between xylose and glucose based on the fact that, in MD simulations, only the combination of D27 protonation and xylose binding significantly destabilize residues in the ligand-binding pocket.

These are indeed the two main messages, and we thank the reviewer for helping us clarify the story.

I have only a few additional comments:

- 1. The authors propose to call the conformational ensemble favored by the combination of D27N and xylose, identified through the HDX-MS experiments, the “transition state.” While I agree that this term is more fitting than their previous term (“active state”), the subsequent HDX-MS signal does not solely arise from the chemical transition state that represents the energetic barrier between two states. Rather, the HDX-MS experiments carried out under different conditions represent ensembles, or combinations of states weighted by their populations. I would therefore prefer if the authors used phrases like “[D27N+xylose compared to D27N alone] favors transition-competent conformations, where transition refers to the conformational change that allows the transporter to move between the outward-facing and inward-facing states.”*

We thank the reviewer for this suggestion. In order to find a balance between readability and accuracy we decided to keep the “transition state” terminology but specify that indeed “transition refers to the conformational change that allows the transporter to move between the outward-facing and inward-facing states.”. Furthermore, we do not use this “transition state” terminology to describe the HDX-MS results anymore, but only to describe the combination of the MD and HDX results

- 2. Apologies if the authors included this and I missed it, but is the D27N transporter a physiologically functional transporter? Have the authors or others measured its*

function by liposome uptake assays or survival assays? If this data exists, I might recommend including mention of it in the Discussion, where the authors discuss such mutants.

We thank the reviewer for pointing this out, this information was probably deleted by mistake during the revision process. D27 is critical for function, more specifically for active transport. This has been demonstrated with cell-based uptake assays and counterflow measurements in liposomes[1-3]. We have updated the introduction and discussion sections accordingly.

3. *The new simulation data, focused on differences in residue positions within the substrate-binding pocket, is exciting to see. I have a couple of questions and follow-up points:*

(a) *In Fig. 5, it is a bit difficult to tell whether xylose and glucose start out in the same pose and then xylose reorients when D27 is protonated, but glucose does not, or if xylose-bound and glucose-bound structures show different poses of the bound substrates. I imagine that the former is the case, but please clarify this somewhere in the text (or simply show the 'first' and 'last' frames instead of the entire ensemble of substrate positions in Fig. 5 (a-c, part i)--if the snapshots shown in Fig. 6 are in fact representative of those states, then please clarify this as well.*

We have edited the text according to the reviewer suggestions. Both xylose and glucose start in the crystal structure pose, and xylose reorients when D27 is protonated.

(b) *Fig. 6a and Fig. S8a might both suffer from the same labeling confusion: in the former, the blue bars in Fig. 6a refer to a "D27 unprotonated, glucose-bound" condition, but I believe that this simulation has D27 protonated, for a comparison to the d27 protonated, xylose-bound condition? If not, then this label is inconsistent the labels in Fig. 5c. The same labeling issue might be happening in Fig. S8a.*

We thank the reviewer for spotting this mistake. It is "glucose bound – D27 protonated". This has now been corrected.

(c) *When showing the plots to demonstrate differences in dihedral conditions, please clarify whether those plots reflect data pooled across each of the two simulation replicates and indicate whether the simulations' behavior is consistent across the two replicates for a given condition.*

We have edited the text according to the reviewer's suggestion. We have added the following modifications. "Differences in the orientation of N294 and Q168 based on the Chi 1 & Chi 2 dihedral maps generated from the combined trajectory of the two replicates".

(d) *Finally, in the text, please clarify whether the changes in helix 1 conformation, and in the rotameric states of Q168 and N294, observed in simulation resemble these*

regions' conformations in IF crystal structures? Or are these regions' conformations observed in simulation representative of a different conformational state?
These changes are also observed in the IF crystal structures. This is now specified in the main text.

Reviewer #3 (Remarks to the Author):

All my comments were satisfactorily taken into account.

Reference

1. Madej, M.G., et al., *Functional architecture of MFS D-glucose transporters*. Proc Natl Acad Sci U S A, 2014. **111**(7): p. E719-27.
2. Wisedchaisri, G., et al., *Proton-coupled sugar transport in the prototypical major facilitator superfamily protein XylE*. Nat Commun, 2014. **5**: p. 4521.
3. Sun, L., et al., *Crystal structure of a bacterial homologue of glucose transporters GLUT1-4*. Nature, 2012. **490**(7420): p. 361-6.